# A variational technique to estimate snowfall rate from coincident radar, snowflake, and fallspeed observations

Steven J. Cooper[1], Norman B. Wood[2], Tristan S. L'Ecuyer[3]

[1]Department of Atmospheric Sciences, University of Utah, Salt Lake City, UT, USA
[2]Cooperative Institute for Meteorological Satellite Studies, University of Wisconsin-Madison, Madison, WI, USA
[3]Department of Atmospheric and Oceanic Sciences, University of Wisconsin-Madison, Madison, WI, USA

*Correspondence to*: Steven J. Cooper (steve.cooper@utah.edu)

**Abstract.** Estimates of snowfall rate as derived from radar reflectivities alone are non-unique. Different combinations of snowflake microphysical properties and particle fallspeeds can conspire to produce nearly identical snowfall rates for given radar reflectivity signatures. Such ambiguities can result in retrieval uncertainties on the order of 100-200% for individual events. Here, we use observations of particle size distribution (PSD), fallspeed, and snowflake habit from the Multi-Angle Snow Camera (MASC) to constrain estimates of snowfall derived from Ka-band ARM Zenith Radar (KAZR) measurements at the ARM NSA Barrow Climate Facility site. MASC measurements of microphysical properties with uncertainties are introduced into a modified form of the optimal-estimation CloudSat snowfall algorithm (2C-SNOW-PROFILE) via the a priori guess and variance terms. Use of MASC fallspeed, MASC PSD, and CloudSat snow particle model as base assumptions resulted in retrieved total accumulations with a -18% difference relative to nearby National Weather Service observations over five snow events. The average error was 36% for the individual events. Use of different but reasonable combinations of retrieval assumptions resulted in estimated snowfall accumulations with differences ranging from -64% to +94 +122% for the same storm events. Retrieved snowfall rates were particularly sensitive to assumed fallspeed and habit, suggesting that in-situ measurements can help to constrain key snowfall retrieval uncertainties. More accurate knowledge of these properties dependent upon location and meteorological conditions should help refine and improve ground and space-based radar estimates of snowfall.

## 1 Introduction

The high-latitude regions play a critical role in shaping climate response to anthropogenic forcing. Model predictions suggest that it is these areas that are most susceptible to change and will experience the most dramatic temperature increase in response to the release of greenhouse gases into the atmosphere (Manabe and Stouffer, 1980; Holland and Bitz, 2003; Serreze and Francis, 2006). Observed changes in the Arctic over the late 20th century and early 21st century have been dramatic and have included increased surface temperature and decreased sea ice, permafrost, glacial ice sheet, and spring

arctic snow cover extents (Serreze *et al.*, 2000; Frauenfeld *et al*., 2004; Dyurgerov and Meier, 2005; Stroeve *et al*., 2008, Brown et al., 2010; Stroeve et al., 2014, Perlwitz et al., 2015, among numerous others).

Snowfall can dramatically change surface conditions at high-latitude regions, acting to increase shortwave surface albedo
while impacting sensible and latent heat fluxes and longwave emission (Cohen and Rind, 1991). Changes in snow cover can feed back on snow cover, sea ice, and permafrost distributions (Brown, 2000; Ramonovsky, 2002; Holland et al., 2006; Vavrus, 2007), where the effects on permafrost in turn may affect high-latitude carbon storage. Snowfall changes at higher north latitudes may also impact freshening of the North Atlantic Ocean and the strength of the Atlantic Meridional Overturning Circulation. Peterson et al. (2006) attributed more than half of the cumulative freshwater input anomaly in the
Arctic and high-latitude North Atlantic Oceans over the previous fifty years to increases in net precipitation over land and oceans, exceeding the estimated contributions from glacial melt and sea ice reduction. Snowfall is also important as it provides mass influx for vast glacial ice sheets such as those found in Antarctica and Greenland (Lenaerts et al., 2013; Palerme et al., 2014). The distribution of snowfall helps define ice sheet dynamics, and the relative difference between snowfall and melt will impact the long-term survival of these glaciers (Van Tricht et al., 2016). Significant loss of ice sheets
may have deleterious effects on human society through a corresponding rise of global ocean levels. (Gardner et al., 2013; Jacob et al., 2013)

The quantitative estimation of snowfall at the global scale from spaceborne measurements has occurred only recently. Initial retrieval approaches were based on passive microwave measurements (Skofronick-Jackson et al., 2004; Noh et al. 2006)
with a shift in emphasis to radar observations with the launch of the CloudSat Cloud Profiling Radar (CPR) in 2006. Matrosov et al. (2007b) and Liu (2008a) demonstrated the first-order capability of the CloudSat CPR to retrieve vertical profiles of dry snowfall. Kulie and Bennartz (2009), in turn, used CloudSat CPR reflectivities to estimate global dry snowfall rate for a year of CloudSat data. They found that snowfall estimates depended critically upon assumed relationships between radar reflectivity and snowfall particle size distributions and shapes.

These studies suggest that estimates of snowfall as derived from radar reflectivities alone are non-unique. Numerous different combinations of snowflake microphysical properties and snow particle fallspeeds may yield nearly identical surface snowfall rates for a given reflectivity profile. As such, use of traditional $Z_e$-S relationships to quantify snowfall cannot be expected to produce accurate results for all snowfall events.

From an operational retrieval perspective, clever selection of representative snowflake microphysical properties may produce estimates of snowfall amounts that agree well with reported values for climate or regional applications. For example, Wood et al. (2015, but see also Wood, 2011) developed snowflake models for the CloudSat snowfall retrieval algorithm based upon field campaigns focused on cold-season clouds and precipitation, principally the Canadian CloudSat-CALIPSO Validation

Project (C3VP, Hudak et al. 2006). They used this a priori knowledge of snowfall microphysics to refine expected snowfall-radar reflectivity relationships for the optimal-estimation (Rodgers, 2000) based CloudSat snowfall retrieval scheme used for the CloudSat 2C-SNOW-PROFILE product. The product provides estimates of snow size distribution and bulk properties (water content and snowfall rate) at the surface and aloft over land, ice and ocean surfaces. These estimates have proven

valuable for assessing snowfall budgets in remote regions (Palerme et al. 2014; Kulie et al., 2016), providing data for testing global climate models (Palerme et al., 2016; Christensen et al., 2016), and evaluating the performance of ground-based radar measurements of snowfall (Norin et al., 2015). The Global Precipitation Mission (GPM) continues the global monitoring of snowfall with duo-frequency Ka-band and Ku-band radar and extensive ground measurement activities (Skofronick-Jackson et al., 2015).

Despite such efforts, validation activities suggest that uncertainties in retrieved snowfall rates can still be on the order of 200% for individual snow events (Wood, 2011). In our retrieval scheme, instead of using a priori guesses of snowflake microphysical properties from field campaigns such as C3VP, we used co-incident observations of snowflake microphysical properties from the Multi-Angle Snow Camera (MASC, Garrett et al., 2012) to constrain the radar-based retrieval approach.

The MASC takes multiple images of snowflakes in free fall while simultaneously measuring fallspeed. From these pictures, estimates of snowflake maximum particle dimension, habit, and other properties that could be used to refine a radar retrieval scheme are estimated. It should be noted, however, that any instrument that measures snow particle properties or fallspeed, e.g. the Precipitation Imaging Package (PIP; Newman et al., 2009), could replace the role of the MASC in the variational approach presented in this work.

We describe here a retrieval technique for snowfall rate and its application to five snow events as observed at the Atmospheric Radiation Measurement (ARM) North Slope Alaska Climate Facility Site at Barrow. Our scheme uses ground-based Ka-band ARM Zenith Radar (KAZR) measurements for reflectivity profiles and was directly modified from the W-band CloudSat 2C-SNOW-PROFILE algorithm. The flexible optimal-estimation framework was used to incorporate

co-incident MASC observations into the radar-based scheme through the retrieval a priori terms. A primary objective was to identify those combinations of retrieval assumptions that allowed the best match with snowfall observations at the Barrow site for five snow events during April and May 2014. Another objective was to quantify the sensitivity of snowfall rate retrieval results to the key microphysical assumptions of particle size distribution, habit, and fallspeed from the MASC. These results were contrasted with snowfall rates found using alternate assumptions such as Locatelli and Hobbs (1974,

hereafter LH74) particle dimension- fallspeed parameterizations, observed KAZR Doppler velocities, and previous field campaign estimates of snow particle size distributions. In Section 2, we discuss the methodology including the CloudSat snowfall retrieval scheme, the MASC observations, and the combined radar- MASC retrieval approach. In Section 3, we explore the application of the combined scheme to five snowfall events observed at the Barrow site. In section 4, we discuss

the implications of our results for the utility of ground-based in-situ measurements to refine and improve radar retrievals of snowfall.

## 2 Methodology

### 2.1 CloudSat Snowfall Retrieval Scheme

The CloudSat snowfall retrieval algorithm uses profiles of W-band 94 GHz radar reflectivity to estimate vertical profiles of snow properties. The flexible optimal-estimation approach is used to combine measurements and a priori microphysical information into a common retrieval cost function. Specifically, the scheme assumes an exponential form for snow particle size distributions for each radar reflectivity bin as in Eq. (1),

$N(D)=N_0\exp(-\lambda D),$           (1)

where $\lambda$ is PSD slope ~~parameter~~, $N_0$ is its intercept, and D is particle maximum dimension (Heymsfield et al., 2008). For our implementation, the PSD slope parameter is allowed to vary with height but the number density is held constant given the limited number of independent observations from the radar. (Note that the uniform $N_0$ used in this work represents a
divergence from the current CloudSat algorithm in which number density is allowed to vary with height.) A priori assumptions of particle mass-diameter and diameter- fallspeed relationships for D in Eq. (1) allow the determination of snow water content and snowfall rate for each radar range bin.

For the CloudSat algorithm, the required a priori microphysical and scattering properties were determined from analyses of
snow observations from field campaigns focused on cold-season clouds and precipitation, principally the Canadian CloudSat-CALIPSO Validation Project (C3VP, Hudak et al. 2006). Snowflake particle models were constructed based upon observed mass and horizontally-projected area as a function of particle size (Wood et al., 2015). These efforts were somewhat unique in that they explicitly use ground-based observations to refine forward model assumptions for the retrieval, an idea we will return to for our combined radar-MASC scheme as described below. Scattering properties for the snowflake
particle models were developed through use of the Discrete Dipole Approximation (DDA, Draine and Flatau, 1994) method. DDA replaces the true particle shape with an approximate shape constructed as a three-dimensional array of small, cubic ice dipoles.

Figure 1 shows the particle models developed for the CloudSat algorithm. These particle models, intended to simulate the
coarse features of snow particles, consist of solid-ice dipoles intermixed with empty (i.e., air-occupied) dipole locations to meet observed mass and horizontally-projected area (Ap) constraints. Analyses of disdrometer, X-band radar, 2D video disdrometer and Snow Video Imager observations for four significant C3VP snow events provided best-estimate mass-

dimension and Ap-dimension relationships (Wood et al., 2015). Scattering calculations for a range of shapes conforming to these constraints suggested that backscatter cross-sections can vary by two orders of magnitude between these models for a given particle size typical of snowfall.

For the C3VP snow events, these particle models were evaluated by testing against coincident, observed W-band radar reflectivities (Wood et al., 2015). DDA scattering properties were used in combination with video disdrometer observations of snow particle size distributions (the Snowflake Video Imager, Newman et al., 2009) to calculate synthetic W-band reflectivities. These synthetic reflectivities were then plotted versus observed reflectivities as in Figure 2. Minimization of these differences suggested that the B8pr-30 model best matched observations for the C3VP snow events.

For our study, we will use this B8pr-30 particle model as a base assumption for our Barrow storm events. Given the non-uniqueness of the snowfall retrieval problem, however, it is possible that sources of error other than particle model may have compensated to allow observed and synthetic reflectivities to match as in Figure 2. We will examine the sensitivity of retrieved snowfall rate to MASC observations of shape, particle dimension, and fallspeed to better understand this issue as

discussed in Section 2.3 below.

**2.2 Multi-Angle Snowflake Camera**

The Multi-Angle Snowflake Camera (MASC) takes high-resolution multi-angle photographs of snowflakes as they settle near the surface. It simultaneously measures snowflake fallspeed. The MASC consists of three cameras each pointing at an identical focal point approximately 10 cm away in a ring opening. This ring houses a system of near-infrared emitter-

detector pairs, arranged in two arrays that are separated vertically by about 3.2 cm. If a hydrometeor passes through both upper and lower arrays, the MASC will trigger each of the three cameras and flash a bank of lights aimed at the center of the camera depth of field. Fallspeed is calculated from the time it takes to traverse the distance between the upper and lower triggering array.

Given images from the MASC, it is possible to derive estimates of properties such as particle shape, aspect ratio, maximum dimension, complexity, and orientation (Garrett et al., 2012; Garrett and Yuter, 2014; Garrett et al., 2015). Here, specifically, we use observations of maximum dimension and habit to constrain uncertainties inherent to radar-based retrievals of snow rate. For example, Figure 3 shows typical snowflakes observed at the NSA Barrow Climate Facility Site during the Spring 2014 MASC deployment. Observed habits include graupels, columns, plates, and aggregate combinations

of each. Such observations of shape can be used to select the most appropriate particle model for a given precipitation scene or to identify those scenes in which none of the currently available particle models will be expected to produce a good fit.

Observations of fallspeed from the MASC, in turn, can be used to translate retrieved cloud snow water content to a snowfall rate. In this work, we contrast retrieved snowfall rates found using MASC observations, near-surface Doppler measurements, and LH74 particle dimension- fallspeed relationships.

### 2.3 Combined Radar- MASC Retrieval

The MASC deployment to the ARM NSA Barrow Climate Facility Site in Spring 2014 provided an ideal opportunity to employ our combined radar-MASC snowfall retrieval scheme. ~~Given the lack of W-band radar measurements during this time period, we use Ka-band ARM Zenith radar (KAZR) general mode observations for this study.~~ In this section, we present sample observations from a Barrow snowstorm to illustrate how we merge MASC information into our radar-based retrieval scheme. We focus specifically on MASC observations of particle maximum dimension, particle model, and

fallspeed. The MASC ideally should be wind-shielded to help reduce sampling errors due to wind and turbulence (Goodison et al., 1998). Given the temporary nature of the deployment at Barrow, however, the MASC was left unshielded during the storm events presented in this work.

Given the lack of W-band radar measurements during this time period, we use Ka-band ARM Zenith radar (KAZR) general

mode observations for this study. The KAZR is vertically-pointing radar that measures reflectivity, vertical velocity, and spectral width at 34.8 GHz with a resolution of 30 m from near surface up to about 20 km (Bharadway et al., 2013). The radar had a 2m antennae with 0.31 degree beam width. KAZR systems have a minimum detectable signal in general mode near -20 dBZ to -25 dBZ dependent upon target range (Feng et al., 2014; Chandra et al., 2015). Such performance is sufficient for these studies given we need reflectivities near -10 dBZ during the entire day to generate measureable snowfall

(0.01 inches) as reported by the National Weather Service. Figure 4 shows KAZR reflectivities for an all-day April 23 snow event at Barrow. The MASC images shown below the reflectivity plot indicate typical observed snowflakes selected as a function of time. For example, these images suggest rimed graupel-like particles around 4 UTC when the KAZR reflectivities suggest the most cloud vertical development. The snowflakes transitioned to sector plate type crystals and aggregates as the cloud tops lowered and became more homogenous.

To understand precisely how we incorporate desired MASC information into the modified KAZR snowfall scheme, it is useful to consider the optimal-estimation approach used in the CloudSat retrieval scheme. Letting **x** denote the vector of snowfall properties to be retrieved, the optimal estimation technique consists of minimizing a combination of the variance between the set of observations, **y**, and a corresponding set of simulated measurements, F(**x**), and between **x** and a suitable a

priori guess, **a**. Assuming Gaussian statistics, this is accomplished by minimizing the scalar cost function,

$$\Phi(x,y,a) = (y - F(x))^T S_y^{-1}(y - F(x)) + (x - a)^T S_a^{-1}(x - a) \qquad (2)$$

with respect to **x**. F denotes the physical model relating snowfall parameters to the radar observations and is called the ''forward model'', $\mathbf{S_a}$ is the a priori error covariance matrix and $\mathbf{S_y}$ is the measurement error covariance matrix. $\mathbf{S_y}$ represents not only random instrument noise but also the impact of uncertainties in forward model assumptions on simulated

measurements, F(**x**). We use a $\mathbf{S_y}$ standard deviation value of 2 dBZ for this study for the diagonal matrix elements based upon Hammonds et al. (2014) who quantified the uncertainty in forward modeled radar reflectivity due to assumptions such as mass-dimensional relationships at Ka-band frequency. The optimal-estimation approach weights the magnitude of the error covariances to determine the relative impact of both a priori and observations on the final retrieval estimates. The observation vector, **y**, is the vertical profile of KAZR reflectivities where the retrieval vector, **x**, would be the vertically-

varying snowfall particle PSD slope parameter and vertically-uniform number density. The values of **x** that minimize Eq. (2) are found through Newtonian iteration.

We introduce MASC PSD information into the retrieval scheme through use of the a priori estimate, **a**, and a priori covariance matrix, $\mathbf{S_a}$. For the PSD slope parameter a priori guess, ~~we processed the MASC images to quantify maximum~~

~~particle dimension for each snowflake~~, the MASC images were processed to quantify maximum particle dimension for each snowflake according to the techniques developed in Garrett et al. (2012), Garrett and Yuter (2014), and Garrett et al. (2015). We then fit a slope parameter for an assumed exponential particle size distribution to the tail of the size distribution > 1 mm in size. We derived estimates of slope parameter either for the entire storm event or for subsections of the event based upon differences in storm morphology. For example, for the April 23 event, we found separate estimates of the slope parameter

corresponding to the vertically developed part of the storm before 8 UTC (1.11 mm$^{-1}$) and for the stratiform part of the storm after 8 UTC (0.74 mm$^{-1}$). The use of such lengthy time periods was necessary given the infrequent sampling of snowflakes by the MASC at the Barrow site, typically from one to ten snowflakes per minute for these events.

Values for uncertainties in the slope parameter as defined in the a priori error covariance matrix were calculated for the

individual storm subsections. These values depended upon both the measure of fit of the exponential slope parameter to MASC observations and expected uncertainties in MASC-derived estimates of particle size from snowflake images. Uncertainties in the slope fit, $\sigma_{\mathbf{FIT}}$, were defined as regression 95% confidence values. Uncertainties in MASC estimates of particle size, $\sigma_{\mathbf{SIZE}}$, were assumed to be 15% of the median maximum particle dimension based upon the preliminary work of Kleinkort et al. (2016). That study used 3-D printed synthetic snowflakes with known geometry to evaluate MASC

estimates of particle dimensions using both a 3-camera and 5-camera system. For our study, the variance of the slope parameter guess, $\sigma_{\mathbf{a}}^{\mathbf{2}}$, is then defined as

$$\sigma_a^2 = \sigma_{FIT}^2 + \sigma_{SIZE}^2 . \qquad (3)$$

Table 1 lists estimated slope parameters with uncertainties for the five storm events presented in Section 3. The relatively small uncertainties in the MASC slope parameter observations dictate a solution close to the a priori guess. As a consequence, the uncertainty in the number density was then allowed to vary over several orders of magnitude to ensure retrieval convergence, i.e. allow the particle number to vary so that the forward model simulated reflectivities can match observed radar reflectivities regardless of particle size. The number density term for this scheme is assumed constant with height. Although this approach may be a problem for space-based radars that must see through the entire storm to estimate surface snowfall rate, it is more applicable for use with the upward-looking KAZR. Retrieved snowfall rates are based only on the lowest non-noise radar range bins from 160-310 m above the surface.

Given a retrieved slope parameter and number density, snow water content was estimated for each radar range bin assuming a knowledge of snow particle mass-dimension relationships. As in Figure 4, MASC images are used to select the most appropriate particle model with associated scattering properties for a given scene. The development of more particle models, e.g. lightly rimed aggregates or graupels to better represent the natural variability of observed snowflakes is a current line of research. In terms of the optimal-estimation technique, this will reduce a major source of uncertainty in the forward model ($S_y$ term in Eq. 2).

Observations of particle fallspeeds from the MASC were used to translate retrieved snow water contents into surface snowfall rates. Figure 5 shows MASC observed snowflake fallspeeds plotted as hourly averages for the April 23 snow event. This plot shows that fallspeeds from the MASC observations were almost always less than those measured from the KAZR Doppler and those predicted from LH74 parameterization schemes (graupel-like snow and aggregates of dendrites). Discrepancies in fallspeed between the MASC and KAZR Doppler observations were expected. Both of these measurements depend upon still air particle fallspeed relationships. But they also depend upon atmospheric vertical motion which is expected to vary between the near-surface MASC and aloft Doppler measurements. The MASC observations also would be influenced by near-surface turbulence and disruptions to airflow as the snowflakes pass through the sampling ring and infrared sensors. Indeed, the hourly MASC and KAZR Doppler fallspeeds had a Pearson correlation coefficient of -0.34. We note that these discrepancies could have been exacerbated, in part, from the unshielded status of the MASC.

The LH74 fallspeeds (m/s) were estimated through equations 4 and 5 as below,

$$V = 1.1\, D^{0.28} \qquad \text{for Graupel,} \qquad (4)$$

$$V = 0.8\, D^{0.16} \qquad \text{for Dendrites,} \qquad (5)$$

where the snowflake maximum dimensions (mm) used for the LH74 parameterization schemes were calculated using MASC observations. The discrepancies in Figure 5 are in general agreement with Garrett and Yuter (2014) who noticed differences between MASC observations of fallspeed at Alta Ski Resort in Utah and LH74 parameterizations. The LH74 schemes predicted fallspeeds poorly for low temperature and highly turbulent environments that might be expected during high-

latitude Arctic snow events. However, the hourly MASC observations still had a higher Pearson correlation coefficient with the LH74 schemes (0.66) than with the Doppler fallspeeds (-0.34).

### 3. Snowfall Retrieval Results

### 3.1 Snow Events and Retrieval Assumptions

The KAZR- MASC retrieval approach was applied to five snow events as observed at the Barrow site during Spring 2014.

KAZR reflectivities for these storms are shown in Figures 4 and 6. These events were selected as they produced measureable snowfall at the nearby Barrow National Weather Service site and triggered co-incident MASC snowflake images. Retrieved estimates of total snowfall accumulation were compared with NWS snowfall observations to evaluate retrieval performance.

We examined the impact of snowflake habit, slope parameter ($\lambda$), and fallspeed assumptions on retrieved snowfall liquid water equivalent for these events. In terms of habit, we selected the CloudSat B8pr-30 particle model, sector plates, and hexagonal columns. The CloudSat particle model not only performed well in CloudSat snowfall validation studies but also was visually consistent with MASC images of general shape of snowflakes seen at Barrow. Sector plates and hexagonal columns were chosen as they too were observed during these snow events. Scattering properties for these pristine habits

were derived following the constrained discrete dipole modeling method described by Wood et al. (2015) with particle dimension relationships from Auer & Veal (1970) and particle mass constraints from Mitchell (1996).

For PSD assumptions, we used MASC estimates of slope parameter ~~($\lambda$)~~ with uncertainties ~~as defined through a priori and a priori covariance terms in Equation 2 and Table 1~~ as listed in Table 1 as introduced through the a priori guess and covariance

terms in Eq. 2. We also employed an a priori PSD $\lambda$ assumption (2.8 mm$^{-1}$) as derived from snowflake observations during the C3VP snow measurement field campaign (Wood et al., 2013) for snow events with similar snowfall rates as the Barrow events. ~~These results suggest that use of the C3VP PSD $\lambda$ provides a reasonable alternative when lacking co-incident MASC measurements.~~ For fallspeed assumptions, we used MASC and KAZR Doppler observations, LH74 fallspeed parameterizations, and a general 1 m/s value. For the MASC fallspeed calculations, retrieved snowfall water contents for a

given KAZR reflectivity profile were converted to precipitation rate via their corresponding average hourly fallspeed value.

We performed retrievals with permutations of habit, PSD, and fallspeed assumptions to determine which combinations produce the best match with snowfall rates observed at the Barrow NWS site. Our best guess or base assumptions were the CloudSat B8pr-30 particle model, MASC fallspeed, and MASC PSD λ. Given the non-unique nature of snowfall retrievals and the difference in location between radar and the NWS observations, we do not pretend that agreement between retrieval results and NWS snowfall observations validates our retrieval assumptions. ~~Furthermore, gauge measurements of snowfall amount suffer wind and temperature dependent uncertainties up to at least 25% (Wolff et al., 2014) and cannot be taken at absolute face value~~. Furthermore, the accurate measurement of snowfall amount from precipitation gauges is a challenging research topic in its own right (Goodison et al., 1998; Rasmussen et al., 2012). Numerous studies over the past century have sought to quantify uncertainties in these measurements and then provide a correction factor for environmental conditions (e.g. Black, 1956; Larson and Peck, 1974; Yang et al., 1995; Yang et al., 2005). During windy and turbulent conditions, for example, the gauges likely will underestimate snowfall amount as the snowflakes cannot settle into the gauge opening. Overall, collection efficiency is a complex function of instrument design, shielding, wind, turbulence, temperature, and topography among other factors (Folland, 1988; Goodison et al., 1998; Rasmussen et al., 2012, Theriault et al., 2015) and can result in snowfall uncertainties on the order of 25-50% over the course of a winter season (Wolff et al., 2014). As such, we understand that the reported NWS snowfall values cannot be taken at absolute face value. However, the numerical experiments presented in this work do quantify the sensitivity of snowfall retrieval results to snow microphysical parameters that can be observed by in-situ instrumentation. A better understanding of the impact of these parameters, in turn, should provide a platform in which we can then examine other sources of error in the snowfall problem.

**3.2 April 23 Snow Event**

Radar reflectivities and MASC images for an April 23 snow event that produced a total snow accumulation of 4.57 mm (0.18 inches) liquid equivalent are shown in Figure 4. Retrieval results assuming different particle model, slope parameter, and fallspeed combinations as described in section 3.1 are presented in Table 2. The percentage difference term in the table is defined through Eq. (6),

$$\% \, Difference = \frac{(Retrieved \, Snowfall - NWS \, Snowfall)}{NWS \, Snowfall} * 100. \qquad (6)$$

Error values ranged from -51% to +132% for this snow event. We found the best agreement with the NWS observations using our 'base' assumptions of MASC fallspeed and PSD λ, CloudSat B8pr-30 particle model with a retrieved accumulated snowfall of 4.88 mm liquid equivalent, a difference of +7%. This accuracy was closely matched with use of the combination of hexagonal columns, KAZR Doppler fallspeeds, and the PSD λ from the MASC. For these two scenarios, the highly

reflective per unit mass hexagonal columns and high fallspeed Doppler observations offset the lower reflective per unit mass CloudSat particle model and low fallspeed MASC observations to produce nearly equivalent snow accumulations. Since hexagonal columns were not observed during the snowstorm, however, that solution is not valid. Worst agreement (+132%) arose from use of the MASC PSD λ, the CloudSat B8pr-30 particle model, and the LH74 graupel fallspeed. This large
discrepancy was driven by the high fallspeed of the LH74 graupel parameterization as seen in Figure 5.

These calculations allowed us to quantify the sensitivity of retrieval results to assumptions of PSD λ, fallspeed, and crystal habit for a given storm event. Assumed PSD slope parameter had the least impact on variability in estimated snowfall accumulation. Snowfall totals of 4.88 mm and 6.07 mm were found when using the MASC PSD λ and the C3VP field
campaign observed PSD λ, respectively, holding fallspeed and habit fixed. In terms of the optimal-estimation scheme, changes in the PSD λ are offset by corresponding changes in the PSD number density. So, larger particles (smaller slope parameter) require fewer particles to match radar reflectivity, and vice versa. Such a relationship modulates the impact of changes in particle size distribution on estimated snowfall rates. These results suggest that use of the C3VP PSD λ provides a reasonable alternative when lacking co-incident MASC measurements.

By contrast, differing assumptions of fallspeed led to a factor of 2 differences in retrieved snowfall rates. Accumulations varied from 4.88 mm with MASC fallspeed observations to 10.62 mm with LH74 graupel-like snow parameterizations, holding base habit and PSD fixed. These results were driven by the variability in fallspeeds as plotted in Figure 5 and the linear impact of fallspeed on retrieved snowfall rate. Interestingly, retrieved snowfall rates using low level Doppler
fallspeeds were twice those found using MASC fallspeed observations. Such results imply the need for further research to determine which or either fallspeed observation is appropriate for a given scene.

Differing assumptions of particle model also led to a factor of 2 differences in retrieved snowfall rates. Estimated snowfall accumulations varied from 4.88 mm for the CloudSat B8pr-30 particle model to 2.23 mm for hexagonal columns, given
fixed fallspeed and PSD assumptions. NWS 'truth' (4.57 mm) fell between results assuming the branched CloudSat particle (4.88 inches) and sector plates (3.33 mm) where both of these habits were seen during the storm event as in Figure 4. Given the presence of rimed particles throughout the storm, however, we would not expect perfect agreement given use of non-rimed particle models for the retrieval.

In addition to total accumulation, we also considered the temporal distribution of the snowfall during the event. Figure 7 plots NWS hourly snowfall accumulations for the April 23 storm against retrieved values arising from different assumption permutations. Estimates from the CloudSat B8pr-30 particle model (with MASC fallspeed and MASC PSD) agreed reasonably well in trend and magnitude with the NWS hourly observations with a Pearson correlation coefficient of 0.65 and

a +7% overestimate of snowfall over the entire event. Retrieval results had a higher correlation with the hourly observations for the first part of the storm (0.92) than the second (0.12) even though differences in subsection snowfall totals were greater for the first section (+11%) than for the second (+1%). However, we do not necessarily expect perfect correlation between the NWS observations and the retrieved values given the differences in location between the snow gauge and KAZR. For example, the NWS snow gauges observed significant snowfall (0.76 mm) over the last two hours of the event even though low KAZR reflectivities suggested perhaps the weakest part of the storm.

The sector plate model (with MASC fallspeed and MASC PSD) produced slightly too little snowfall throughout the course of the event. The hexagonal model (with MASC fallspeed and MASC PSD) produced even less. The three curves for the retrieval scenarios using the MASC fallspeeds in Figure 7 were highly correlated with each other (coefficients of 0.99) given common reflectivity and fallspeed assumptions. These curves had a weaker correlation (coefficients near 0.8) with the Doppler scenario given marked discrepancies in fallspeed as seen in Figure 5.

### 3.3 May 15 Snow Event

Table 3 lists retrieved accumulated snowfall amounts for a May 15 snow event as depicted in the upper left panel of Figure 6. This storm produced approximately 7.62 mm (0.30 inches) liquid equivalent of snowfall as observed by the NWS before transitioning to light rainfall. Although the use of our base CloudSat particle model, MASC fallspeed, and MASC $\lambda$ assumptions produced snowfall estimates close to NWS observations for the April 23 event, these same assumptions performed poorly for this storm. We found a -43% difference relative to NWS observations.

For all retrieval permutations, differences between retrieved snowfall and NWS observations varied from -74% to +56%. The best agreement with NWS values came with the permutations of sector plates, MASC $\lambda$, and KAZR Doppler fallspeed assumptions (+6%) and CloudSat B8pr-30 particle model, MASC $\lambda$, and LH74 dendritic aggregate fallspeed assumptions (-10%). Again, it is the compensating nature of fallspeed and particle model for these two different retrieval permutations that allow them to produce results consistent with NWS observations. Agreement was particularly poor for those cases that assumed the highly reflective hexagonal column particle model, e.g. a difference of -74% was found with columns when used with MASC observed fallspeeds and PSD $\lambda$. The use of these columns (which were not seen during the event) could not generate enough retrieved snowfall to match observations regardless of fallspeed assumptions.

A possible reason for these observed discrepancies in this case is the presence of heavy riming in observed snowflakes. MASC images suggest lumpy graupel-like snowflakes and aggregates throughout the entire snow event as shown in Figure 8. Such particles would be expected to be denser and thus have more mass for given maximum dimension than the un-rimed particle models employed in this study. Such conditions would cause the snowfall retrievals to underestimate snow rate, all

other variables fixed. Similarly, particle- backscatter relationship would change given a coating of frozen or possibly liquid water on the particles, further biasing results. Current work by the authors focuses on scattering calculations for rimed particle models assuming different densities. Ideally, these properties could be used with a hydrometeor classification scheme that estimates degree of riming and melt water such as Praz et al. (2017) or similar for better retrieval performance.

**3.4 Totals for Five Snow Events**

Table 4 lists total retrieved liquid water equivalent over the five snow events at Barrow for different retrieval assumption permutations. The approximate accumulated snowfall was 16.0 mm (0.63 inches) liquid equivalent for these five events as measured by the nearby NWS site. Use of our base assumptions (MASC fallspeed, MASC PSD $\lambda$, and CloudSat particle model) led to total accumulated snowfall of 13.11 mm, which represents a difference of -18% relative to NWS observations.

Use of the C3VP a priori PSD $\lambda$ led to a slightly better match with NWS observations (-3%) than use of MASC PSD $\lambda$ (-18%) when used with base particle model and fallspeed assumptions. This relatively close agreement in results is not surprising given the limited sensitivity of retrieval results to changes in PSD $\lambda$. Also, the a priori $\lambda$ value of 2.8 mm$^{-1}$ falls within the range of observed MASC PSD $\lambda$ values from 0.74 to 3.42 mm$^{-1}$. ~~The average error for the individual snow events, however, was similar for the two $\lambda$ assumptions. Value of 43% and 44% were found for C3VP and MASC PSD, respectively.~~ So, it would be expected that retrieved values using the a priori assumption would be greater than those found using MASC values for some cases but less than for others, producing similar results when totaled over the five events.

Other assumption permutations found retrieved liquid water accumulations with differences ranging from -64% to ~~+94%~~ +122% relative to NWS observations. These results again highlight the likelihood of compensating errors when inverting radar observations to estimate snowfall. Best agreement (3%) came for two retrieval scenarios with different assumptions for each habit, PSD $\lambda$, and fallspeed. The CloudSat particle model, C3VP a priori PSD $\lambda$, and MASC fallspeed combination resulted in 3% less retrieved snowfall than NWS observations. Sector plates, MASC PSD $\lambda$, and a 1 m/s fallspeed resulted in 3% more retrieved snowfall than NWS observations. To further illustrate this idea of compensating uncertainties, consider the use of assumptions that are all demonstrably wrong (hexagonal columns, 1 m/s fallspeed, and C3VP a priori PSD $\lambda$) for the vast majority of these observed snow events. This retrieval permutation generates a value for accumulated snowfall of 11.94 mm that is very close to the 13.11 mm found using our base assumptions. ~~In the next section, we will discuss implications of these results for the utility of ground-based in situ measurements to refine and improve estimates of snowfall from radar measurements.~~

The use of total accumulation over multiple snow events provides a practical metric to evaluate retrieval performance over time. However, it is also important to evaluate the average error for the individual storm events since a retrieval scheme

could produce perfect results over multiple storm events while being markedly wrong for each event. Table 4 lists the accumulation-weighted average absolute differences for the individual snow events for different retrieval permutations. We chose this accumulated-weighted approach since two of the storm events had snowfall accumulation less than 1.05 mm liquid equivalent. If we had equally weighted the events, small differences in retrieved snowfall values on the order of 0.5 mm for these events could heavily skew overall average differences.

Table 4 shows accumulation-weighted average absolute differences ranging from 21% to 122% relative to NWS observations for the different retrieval permutations. For our base assumptions (MASC fallspeed, MASC PSD λ, and CloudSat particle model), we found an accumulation-weighted average absolute difference of 36%. This result is similar to the 35% value found with use of the C3VP a priori PSD λ, again demonstrating the limited sensitivity of retrieval results to changes in PSD λ for these events. Two of the smallest average difference values (21% and 27%) were found using hexagonal columns. Again, since columns were very rarely seen during these events, it is likely that such agreement arises from compensating errors from other retrieval assumptions. Similar average difference values (23%, 27%, and 27%) were found for a variety of retrieval combinations based upon a sector plate particle model. Such consistent results identify the sector plate model as a likely habit candidate for a Barrow based snowfall retrieval scheme.

In theory, analyses of retrieval performance could be broken down as a function of meteorological conditions. For example, we might expect that the retrievals would match snow gauge observations poorly under windy conditions. Snowfall gauge measurements can suffer significant uncertainties that are a strong function of wind speed (Black, 1956; Larson and Peck, 1974; Goodison et al., 2008; Yang et al., 2005, Wolff et al., 2014; among numerous others). Likewise, sampling artifacts for MASC particle size and fallspeed observations are likely as increased wind speed and turbulence disrupt the settling of snowflakes as they pass through the MASC sampling volume. In practice, we had only a handful of snow events during the Barrow deployment, making a thorough exploration of this topic difficult. For example, retrievals with our base assumptions matched the snow gauge observations better for the slightly less windy April 23 event (2.85 m/s) than the May 15 (4.40 m/s) event. For the April 23 event, however, the hourly snow gauge and retrieval estimates were more highly correlated during the windier first part of the storm (3.96 m/s) than the calmer second part (2.17 m/s). Both events were also complicated by habit uncertainties, making it difficult to identify a clear wind signature without more observations of snow storms.

Perhaps of more interest, the May 21 event had the strongest average winds at 9.41 m/s (21.0 mph), the greatest measured slope parameter, and thus the smallest particles. These results would be consistent with blowing snow composed of shattered ice crystals. If so, we would expect poor retrieval performance as the PSD and habit measurements near the surface would not be representative of the snowfall conditions aloft. The total snowfall accumulation for this high wind event was only 1 mm (LWE), so again it would be unwise to make conclusions on retrieval performance until we get more data.

Finally, we contrast snowfall estimates found with our variational approach with those found from two different 35 GHz reflectivity- snowfall relationships. Our use of in-situ observations from the Barrow event, in theory, should produce more accurate results than those derived from these parameterizations. Kulie and Bennartz (2009, hereafter KB09) estimated snowfall as in Eq. 7,

$$Z_e = 24.04 S^{1.51} \quad (7)$$

where they assumed a three-bullet rosette particle model (Liu 2008b) with scattering particles derived from DDA. S is snowfall in mm hr$^{-1}$. Matrosov (2007a, hereafter M07) assumed aggregate snowflakes with scattering properties modeled from spheres and T-matrix theory to parameterize snowfall through Eq. 8.

$$Z_e = 56.00 S^{1.2} \quad (8)$$

These equations were derived for vertically pointing radar and dry snowfall conditions that lack significant attenuation at 35 GHz.

Table 5 lists total estimated snowfall and accumulation-weighted average absolute differences for the KB09 and M07 schemes for the five Barrow storm events. KB09 found a total accumulation with +37% difference relative to NWS observations with an average error of 50% for the individual events. M07 found a total accumulation with -48% difference relative to NWS observations with an average error of 49% for the individual events. These values fall well within the range of results presented in Table 4, although they are slightly larger than those found from the better case optimal-estimation retrieval scenarios. Likewise, the correlation coefficient between hourly $Z_e$-S estimates and NWS observations (0.41) was slightly worse than that found between the base OE scheme and NWS observations (0.65). Again, it is hoped the continued refinement of our combined radar, in-situ approach will produce results consistently more accurate than such $Z_e$-S relationships.

## 4 Discussion and Conclusions

In this work, we present an optimal-estimation retrieval scheme to calculate surface snowfall rates using coincident radar and in-situ snow particle observations. The scheme was modified from the W-band CloudSat snowfall algorithm and applied to measurements from the ground-based Ka-band ARM Zenith Radar (KAZR) located at the ARM NSA Barrow Climate Facility Site. Multi-Angle Snow Camera (MASC) estimates of particle size distribution and fallspeed were used to constrain the inverse calculations based upon KAZR reflectivities. Images of snowflakes from the MASC were used as a guide to select the most appropriate particle model for a given storm event, e.g. branched aggregates, sector plates, or columns.

Retrieved snowfall accumulation were compared with snowfall measurements at the nearby NWS Barrow office for a first order evaluation of our results.

Retrieval snowfall values found using MASC observations as assumptions were contrasted with those found using alternate assumptions such as Locatelli and Hobbs fallspeed parameterizations, Doppler fallspeed observations, and field campaign observations of snow particle size distributions. Use of these different permutations of retrieval assumptions (habit, PSD λ, and fallspeed) allowed us to determine which combination of assumptions best matched nearby NWS snowfall observations for five different snow events. Differences between these approaches also quantified the sensitivity of estimated snowfall amounts to PSD λ, particle fallspeed, and snowflake particle model. Although the number of events and snowfall totals with co-incident MASC and KAZR observations is limited, they do provide an initial data set to demonstrate technique and to quantify retrieval performance across multiple snow events.

Use of the base assumptions (CloudSat particle model, MASC fallspeed, and MASC PSD λ) resulted in estimated snowfall totals over the five events with a -18% difference relative to nearby NWS snowfall observations. ~~The average absolute difference was 45% for the individual events.~~ This agreement, of course, could result in part from compensating errors in individual storm totals. We also calculated the accumulation-weighted average absolute difference for the individual events and found a value of 36%. Such results demonstrate that modification of the CloudSat particle model to Ka-band frequency and use of MASC observations can produce reasonable snowfall values for spring conditions at Barrow. This agrees in spirit with the validation studies from the C3VP Program in which simulated reflectivities using the CloudSat particle model matched well with field observations as demonstrated in Figure 2. The use of the sector plate model with different fallspeed and PSD assumptions often produced average error values less than 30%, suggesting the sector plate as a candidate for a Barrow-based snowfall scheme. These results are not surprising since both aggregates and sector plates were seen during these snow events. These better case retrieval scenarios matched NWS observations more closely than two 35 GHz $Z_e$-S relationships that had average errors near 50% for the Barrow events.

Other combinations of retrieval assumptions found differences in accumulated snowfall ranging from -64% to ~~+94~~ +122% relative to NWS observations as listed in Table 4. The accumulation-weighted average absolute difference values for these permutations ranged from 21% to 122%. The non-unique nature of the snowfall retrieval problem, however, makes it difficult to determine if we get good results for the 'right reasons'. For example, the use of a trio of assumptions (hexagonal column particle model, C3VP a priori PSD λ, and a 1 m/s fallspeed) that are all demonstrably wrong yielded overall differences (-25%) that are similar to those found with our base assumptions (-18%). Further, this combination had an average difference (28%) for the individual storms that was indeed better than the value found for our base assumptions (36%). For this 'wrong' scenario, compensating errors in fallspeed (bias results high) and particle model (bias results low)

likely offset to produce snowfall results that are reasonable across multiple snow events. The use of MASC or other habit observations would eliminate these column scenarios as valid solutions in our retrieval approach.

Use of alternate fallspeed and particle model assumptions led to factor of 2 differences in retrieved snowfall rates when averaged over the five snow events. Assumed PSD slope parameter had less impact on variability in estimated snowfall accumulation. Snowfall totals of 13.11 mm and 15.47 mm were found when using the MASC PSD $\lambda$ and the C3VP field campaign observed PSD $\lambda$, respectively, holding fallspeed and habit fixed. In terms of the optimal-estimation approach, changes in the PSD $\lambda$ are offset by corresponding changes in the particle number density. Larger particles require fewer particles to match radar reflectivity, and vice versa, thus limiting the impact on estimated snow water content with changing particle size.

Significant sensitivities to multiple variables imply difficulties for the design and evaluation of radar-based snowfall retrieval schemes. Those studies that focus on one or even a few variables may ~~vary~~ likely lead to misleading conclusions. ~~Meaningful advancements in the retrieval problem will require a comprehensive approach that considers coincident in situ observations of habit, PSD, fallspeed and other variables such as particle orientation. Incremental improvements in the understanding of any of these parameters dependent upon specific location and meteorological conditions will help reduce retrieval non-uniqueness and should allow insights into the other sources of error for the approach.~~ Again, we hope the use of coincident in-situ observations of snowfall microphysical properties will reduce the non-uniqueness of the retrieval problem.

Future work will focus on expanding the coincident radar and ground-based in-situ instrumentation approach presented here. We will modify existing particle models and DDA scattering properties ~~based upon information from the MASC images. For example, MASC images shown in Figures 4 and 7 suggest some degree of riming for most snowflakes even for the cold Barrow site.~~ to match the graupels and rimed particles often seen in the MASC images as in Figures 4 and 8. Differences in the scattering properties and mass-dimensional relationships for rimed and un-rimed particles are anticipated to explain the large discrepancies in retrieved and observed snowfall rates observed for the rimed May 15 case. Use of a hydrometeor classification scheme that estimates degree of riming from MASC images such as that developed by Praz et al. (2017) would provide a quantitative manner to link snowflake images with particle scattering properties. Likewise, we will seek to determine the most appropriate fallspeed metric given available instrumentation. Large discrepancies were observed between near-surface MASC fallspeed measurements, lowest radar range bin Doppler velocities, and particle fallspeed-dimension parameterizations. Meaningful results from such studies will require the quantification of sampling artifacts for the MASC or other in-situ microphysical instrumentation through efforts similar to Kleinkort et al. (2016). They will also require use of state-of-the-art snow gauge measurements, to the extent possible.

Given the limited data available from the NSA Barrow site, we stress the need for more data sets with coincident radar, snowfall microphysical, and snow gauge observations. The general technique presented here for the KAZR and MASC could be adapted to any set of coincident radar and PSD instrumentation. Along these lines, the authors are deploying a Micro Rain Radar, MASC, and Precipitation Imaging Package (PIP) to two snowfall measurements sites in Scandinavia over the next ~~couple of~~ two winters. These sites, one run by the Norwegian Meteorological Institute near Haukeliseter Fjellstue and one by the ~~Swedish Meteorological and Hydrological Institute near Åre~~ Swedish Institute of Space Physics near Kiruna, experience numerous snowfall events representing a diverse range of synoptic and mesoscale conditions. Such a comprehensive set of observations should allow us to refine our retrieval approach and to gain insights into the state-dependence of snowfall microphysics. In turn, these improved snowfall estimates could be used to explain observed differences between ground-based radar, satellite-based radar, and snow gauge estimates of snowfall (Cao et al., 2014; Smalley et al. 2014; Norin et al., 2015; Saltikoff et al., 2015; Speirs et al., 2017) or as input for weather and climate studies.

## Acknowledgments

Data were obtained from the Atmospheric Radiation Measurement (ARM) Program sponsored by the U.S. Department of Energy, Office of Science, Office of Biological and Environmental Research, Climate and Environmental Sciences Division. All authors were supported through National Science Foundation grant 1531930 and Department of Energy grant DE-SC0016045. In addition, parts of this research conducted by N. Wood and T. L'Ecuyer were performed at the University of Wisconsin- Madison for the Jet Propulsion Laboratory, California Institute of Technology, sponsored by National Aeronautics and Space Administration CloudSat Research Grant G-3969-1. Parts of the original code development by S. Cooper was performed under NASA grant number NNX15AK17G and NSF grant 1303965. We thank Tim Garrett for processing the MASC data.

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

**Table 1:  MASC observed slope parameters for exponential particle size distributions for the Barrow snow events.  The C3VP observations were derived from field observations with similar snow rates as the Barrow storms (Wood et al., 2013).**

| Event | PSD Slope Parameter, $\lambda$ | Uncertainty in $\lambda$ |
|---|---|---|
| April 23   00-08 UTC | 1.11 | $\pm 0.23$ |
| April 23   08-24 UTC | 0.74 | $\pm 0.15$ |
| May 15   04-23 UTC | 1.08 | $\pm 0.25$ |
| May 17   11-18 UTC | 1.44 | $\pm 0.29$ |
| May 17   18-24 UTC | 0.74 | $\pm 0.51$ |
| May 21   20-24 UTC | 3.42 | $\pm 0.64$ |
| May 26   08-17 UTC | 2.09 | $\pm 0.68$ |
| C3VP Field Observations | 2.8 | $\pm 0.5$ |

**Table 2:  Retrieved snowfall amounts for an April 23 snow event for designated retrieval assumption combinations. Nearby NWS observations suggested 4.57 mm (0.18 inches) of snowfall liquid equivalent.  The MASC-derived PSD slope parameters (λ) used for this event are listed in Table 1.**

| Particle Model | λ (PSD slope) | Fallspeed | Snowfall (mm) | % Difference |
|---|---|---|---|---|
| CloudSat | MASC | MASC obs | 4.88 | +7 |
| CloudSat | MASC | Doppler | 9.17 | +100 |
| CloudSat | MASC | LH74, Aggs Den | 7.24 | +58 |
| CloudSat | MASC | LH74, Graupel | 10.62 | +132 |
| CloudSat | MASC | 1 m/s | 8.31 | +82 |
| CloudSat | Field- C3VP | MASC obs | 6.07 | +33 |
| Sector Plates | MASC | MASC obs | 3.33 | -27 |
| Sector Plates | MASC | Doppler | 6.30 | +38 |
| Sector Plates | MASC | 1 m/s | 5.72 | +25 |
| Sector Plates | Field- C3VP | 1 m/s | 6.02 | +32 |
| Hex Columns | MASC | MASC obs | 2.23 | -51 |
| Hex Columns | MASC | Doppler | 4.24 | -7 |

**Table 3: Retrieved snowfall amounts for a May 15 snow event for retrieval assumption combinations. Nearby NWS observations suggested 7.62 mm (0.30 inches) of snowfall liquid equivalent.  The MASC-derived PSD slope parameters (λ) used for this event are listed in Table 1.**

| Particle Model | λ (mm^-1) | Fallspeed | Snowfall (mm) | % Difference |
|---|---|---|---|---|
| CloudSat | MASC | MASC obs | 4.34 | - 43 |
| CloudSat | MASC | Doppler | 11.91 | + 56 |
| CloudSat | MASC | LH74, Aggs Den | 6.88 | - 10 |
| CloudSat | MASC | LH74, Graupel | 9.63 | + 26 |
| CloudSat | MASC | 1 m/s | 8.41 | + 10 |
| CloudSat | Field- C3VP | MASC obs | 5.33 | - 30 |
| Sector Plates | MASC | MASC obs | 2.95 | - 61 |
| Sector Plates | MASC | Doppler | 8.10 | + 6 |
| Sector Plates | MASC | 1 m/s | 5.72 | - 25 |
| Sector Plates | Field- C3VP | 1 m/s | 6.02 | - 21 |
| Hex Columns | MASC | MASC obs | 1.98 | - 74 |
| Hex Columns | MASC | Doppler | 5.49 | - 28 |

**Table 4:** Total retrieved snowfall amounts over five Barrow snow events for designated retrieval assumption combinations. Nearby NWS observations suggested 16 mm (0.63 inches) of snowfall liquid equivalent. The MASC-derived PSD slope parameters used for these events are listed in Table 1.   The '% Difference' value refers to the difference between the NWS and retrieved accumulations over the five snow events.   The 'Avg % Difference' value refers to the average difference for the individual events.

| Particle Model | $\lambda$ (mm^-1) | Fallspeed | Snowfall (mm) | % Difference | Avg % Difference |
|---|---|---|---|---|---|
| CloudSat | MASC | MASC obs | 13.11 | -18 | 36 |
| CloudSat | MASC | Doppler | 30.35 | +89 | 89 |
| CloudSat | MASC | LH74, Aggs Den | 20.60 | +29 | 38 |
| CloudSat | MASC | LH74, Graupel | 30.99 | +94 | 94 |
| CloudSat | MASC | 1 m/s | 25.63 | +60 | 60 |
| CloudSat | Field- C3VP | MASC obs | 15.47 | -3 | 35 |
| CloudSat | Field- C3VP | Doppler | 35.53 | +122 | 122 |
| Sector Plates | MASC | MASC obs | 8.58 | -46 | 46 |
| Sector Plates | MASC | Doppler | 19.74 | +23 | 23 |
| Sector Plates | MASC | 1 m/s | 16.46 | +3 | 27 |
| Sector Plates | Field- C3VP | 1 m/s | 17.20 | +7 | 27 |
| Hex Columns | MASC | MASC obs | 5.79 | -64 | 64 |
| Hex Columns | MASC | Doppler | 13.38 | -16 | 21 |
| Hex Columns | Field- C3VP | 1 m/s | 11.94 | -26 | 28 |

**Table 5:** Retrieved snowfall amounts as derived from two different $Z_e$-S relationships for April 23, May 15, and all snow events seen at Barrow. These results are contrasted with those found using our base optimal-estimation scheme and NWS observations. Difference values are defined as in Table 4.

| | April 23 | | May 15 | | All Events | | |
|---|---|---|---|---|---|---|---|
| | LWE (mm) | % Difference | LWE (mm) | % Difference | LWE (mm) | % Difference | Avg % Diff |
| NWS | 4.57 | _ | 7.62 | - | 16.0 | - | - |
| Base Scheme | 4.88 | +7 | 4.34 | -43 | 13.11 | -18 | 36 |
| Kulie 09 | 8.71 | +90 | 6.63 | -13 | 21.97 | +37 | 50 |
| Matrosov 07 | 3.53 | -23 | 2.68 | -65 | 8.37 | -48 | 49 |

**Figures**

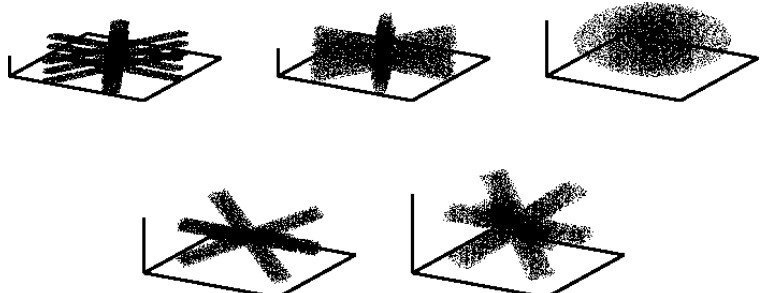

**Figure 1: Discrete dipole models, clockwise from upper left: Sector plate, SPp; 6-branched planar rosette, B6pf; ellipsoid, Ep; 8-branched rosette with 0.5 aspect ratio, B8pr-30; 8-branched rosette with 0.7 aspect ratio, B8pr-45 as taken from Wood et al. (2015).**

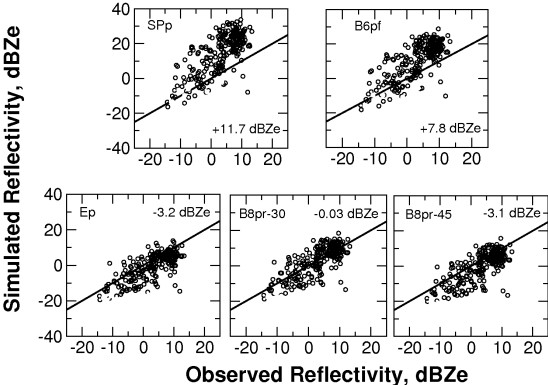

**Figure 2: Comparisons of observed W-band reflectivities with synthetic reflectivities derived from the DDA scattering properties**

15 **as taken from a modified figure from Wood et al. (2015).**

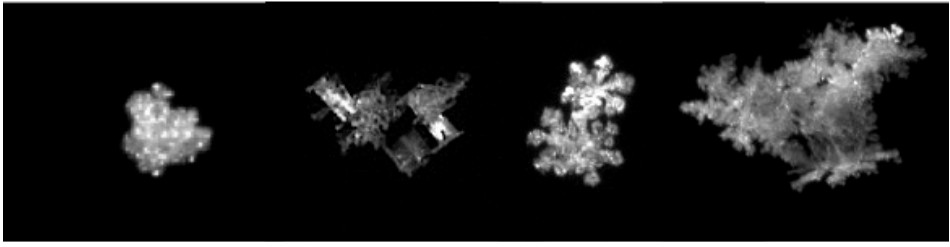

**Figure 3. MASC-captured images of snowflakes as observed at the NSA Barrow Climate Facility site.**

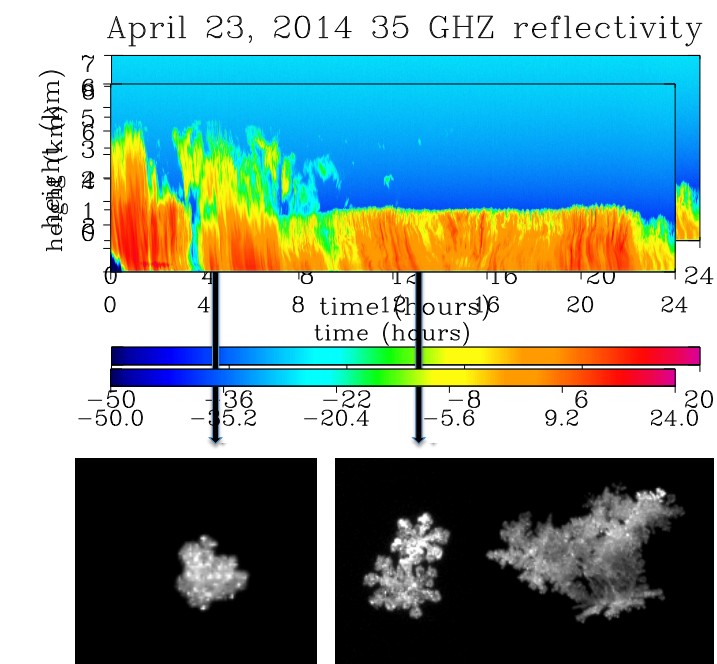

**Figure 4: The top scene shows KAZR reflectivities (dBZ) for an April 23 snow event at the NSA Barrow site. MASC images of snowflakes suggest a graupel-like structure around 4 UTC which transitioned to more pristine structures and aggregates by 13**

10 **UTC.**

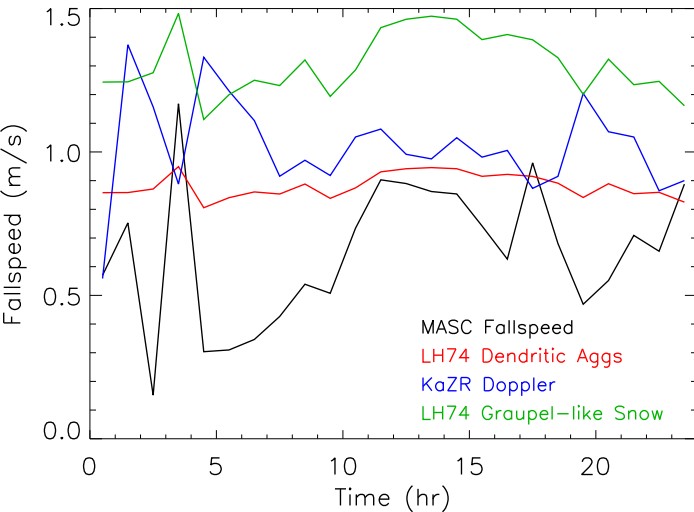

**Figure 5. MASC observed fallspeeds for April 23 snowfall event at Barrow are plotted in black. Results are compared to calculations from LH74 parameterization schemes for crystal types similar to those observed by MASC during the storm event. Green represents graupel-like snow while red represents dendritic aggregates. Near surface Doppler fallspeeds from the KAZR are plotted in blue. The MASC fallspeed observations had correlation coefficients of -0.34 and 0.66 with the KAZR and LH74 fallspeeds, respectively.**

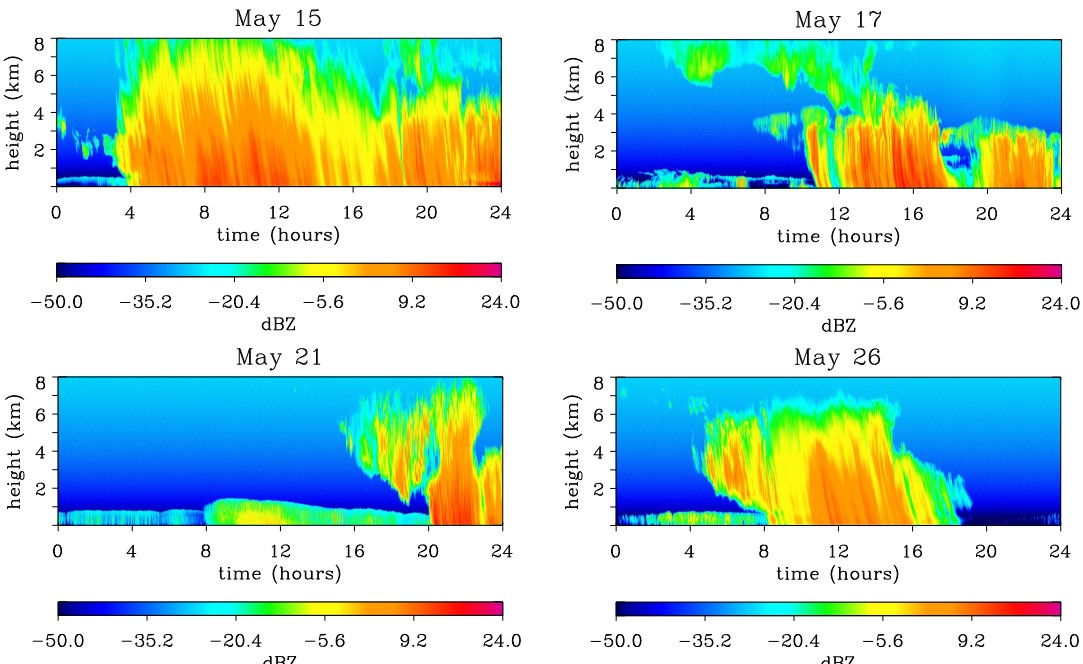

**Figure 6: KAZR reflectivities (dBZ) for snow event systems as observed at the NSA Barrow Climate Facility site. Areas of orange and red indicate likely periods with snowfall.**

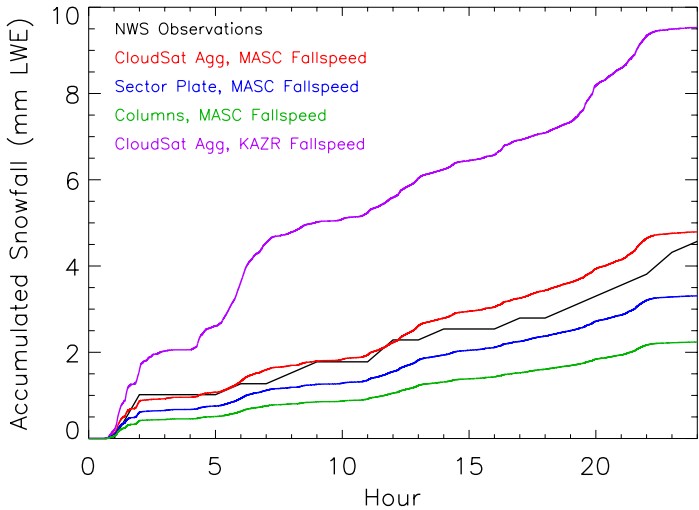

**Figure 7: Hourly NWS snowfall accumulations (mm) plotted against retrieved values for the April 23 event for different retrieval assumption permutations. MASC PSD slope parameter (λ) is assumed for all curves.**

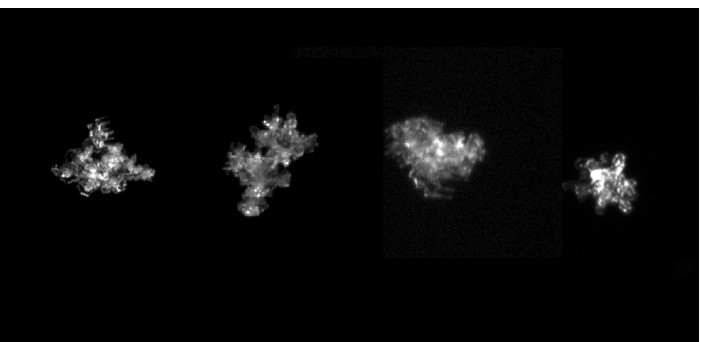

**Figure 8: Typical rimed and graupel-like particles observed by the MASC on May 15.**

