# Peer review of "A variational technique to estimate snowfall rate from coincident radar, snowflake, and fallspeed observations"

_Atmospheric Measurement Techniques, 2017_

## Referee Comment (RC1) · Anonymous Referee #1 · 2 Mar 2017

In this study, the authors utilize MASC measurements of snowflake microphysical properties and particle fallspeeds to constraint a snowfall retrieval scheme applied on Ka-band zenith radar reflectivities. The snowfall amounts thus obtained are then compared with values obtained by using different combinations of retrieval assumptions for the particle model, PSD parametrization and average fallspeed. The paper is well-conceived and presents valuable content. I have two major comments and a few minor comments, mostly clarifications, that should be considered by the authors before publication.

Major comment :

1) In section 3 (results), you emphasize multiple times the non-uniqueness of snowfall

retrievals from radar reflectivities and the issues that it implies: difficulty to assess which combination of retrieval assumptions is truly the best, compensating errors can lead wrong combinations of assumptions to snowfall accumulation in agreement with the reference, etc. The results are presented individually for 2 events and eventually merged into one table showing the total snowfall amount accumulated over five snow events. Given these counterbalancing error effects, I think showing the total snowfall amount error is not very informative and could be misleading. Alternatively, the average error (that you mention p.12, l.8) and its variability are in my opinion more relevant. I would suggest to add this information in Table 3 and discuss the values obtained in Section 3.4 Along the same idea, it would be interesting and relevant to analyze the temporal evolution of snowfall accumulation as retrieved using different retrieval assumptions and compare it with the rain gauge measurement, especially for the April 23 event where you use 2 different estimates of the PSD slope parameter for 2 different periods. For instance, a correlation analysis could provide relevant complementary information to evaluate what are the best combinations of retrieval assumptions and to quantify the sensitivity of the retrieval results.

2) In my opinion, the paragraphs related to fallspeed estimation require some clarifications. How are the fallspeed measured by the MASC inserted into the retrieval scheme? Are you using hourly average as illustrated on Figure 5? How is the fallspeed calculated when multiple particles are present in one MASC image? As you mentioned, Garrett and Yuter 2014 showed that MASC fallspeed measurements are strongly influenced by wind and local turbulence. This will have an impact on the average observed fallspeed but also on its variability. What is the fallspeed variability during the events of interest? You could for instance display this variability on Figure 5 (for MASC and Doppler radar). How was the MASC deployed during the campaign? Was it windshielded? Horizontal wind and turbulence have also a significant impact on the catching ratio of rain gauges. As a rain gauge is used for reference in this study, it would be useful to document the wind conditions during the events and discuss the impact on the results (if available).

Minor comments :

p2, l19-25: As this paragraph is about spaceborne measurements of snowfall, I would suggest to mention the more recent Global Precipitation Measurement (GPM) mission carrying a dual-frequency precipitation radar. Possible citation: Global Precipitation Measurement Cold Season Precipitation Experiment (GCPEX): For Measurement's Sake, Let It Snow, Skofronick-Jackson, Gail, et al. BAMS 2015. p3, l13: I think MASC is the acronym for Multi-Angle Snowflake Camera (not Multiple-Angle Snow Camera). p3, l17: You could add a reference for the Precipitation Imaging Package. p3, l30: we discuss the methodology p4, l10: lambda is the PSD slope [. . .] and D is the particle maximum dimension p4, l24: What do you mean by cubic ice dipoles? Ice dipoles arranged on a cubic grid? p5, l5: I am not sure we can call the SVI a disdrometer, maybe reformulate. P5, l1-14: What assumptions are used on the orientation of the particles in the model? p5, l18 (and others): For a publication in an international journal as AMT, I would suggest to convert all measurements in SI units and potentially keep the Imperial units in brackets. p5, l28: graupels p6, l27: Please clarify why you use this standard deviation value (2 dBZ) p7, l11-18: If I understood correctly, you use MASC-derived PSD slope parameter as a priori guess for the whole profile of KAZR reflectivities. If so, would it make sense to take into account the altitude in the variance of the slope parameter guess (sigma), as the MASC measurement of PSD is less likely to be representative of the real PSD higher in the precipitation column? p7, l28: are based only on the [...] p8, l2: S_y instead of Sy (subscript) p8, l15-16: Please precise the units for V and D. p9, l5: Are sector plates and hexagonal columns models also coming from CloudSat DDA simulations? I see a sector plate model on Figure 1 but not a hexagonal column. Please clarify this point. p9, l10: "a priori" is written two times. p9, l13: Even though C3VP slope parameter lies within the range of values measured by the MASC, it is almost 4 times larger than the measured value for certain events. I would move this sentence to page 10 after you showed that PSD slop parameter has the least impact on the variability in estimated snowfall accumulation. p9, l29: snow accumulation of p11, l24-27: More specifically, rimed snowflakes are expected to be

more reflective, hence reducing the PSD number density for a fixed slope parameter. It may compensate somewhat the increased snowfall retrieval due to the higher density of individual particles. Is this heavy riming also present in the fallspeeds measured by the MASC and the Doppler radar, compared to non rimed events? If so, I would suggest to mention it in this paragraph. P12, l26: Multi-Angle Snowflake Camera p14, l5: Remove "vary" p14, l11-15: In this regard, you may want to cite the recent work by Praz et. al "Solid hydrometeor classification and riming degree estimation from pictures collected with a MASC" (AMTD, 2017) in which the authors developed a method to automatically identify the type of snowflake and the riming degree from MASC images.

---

## Referee Comment (RC2) · Anonymous Referee #2 · 23 Mar 2017

The paper describes how observations from the innovative in situ sensor called Multi-Angle Snowfall Camera (MASC) can be used to constraint a snowfall retrieval scheme from non-Rayleigh echoes detected by a vertically pointing radar working at the attenuating frequency of 35 GHz. The retrieved snowfall amounts are then compared with reference observations during only 5 events, 3 of them weak (total amounts of these 3 events corresponds 3.8 mm snowfall liquid equivalent).

Personally, I feel that the paper has some major points that need to be addressed before it can be considered for publication. I was somehow surprised that the authors do not describe neither the in situ snowfall measurements device nor the remotely

sensed observations (the radar). I would suggest

1] to complement the Mean Error with (at least one or two) complementary additional score, for instance, the standard deviation of the error and the correlation coefficient among hourly amounts. Indeed, it is interesting and relevant to have an idea of the temporal evolution of the recorded snowfall amounts during a period of time shorter than the whole event.

2] Three events out of five are indeed weak. Would it be possible to please add more events?

3] Would you please supply more information regarding the reference in situ observations? Furthermore, please write a few sentences characterizing the mountainous site at hand. In addition to snowfall amounts and water equivalent, do you have information regarding the wind intensity (and direction)? All considered, after having read the interesting the cited paper by Wolff et al. (2014) and other "grey" literature (extended abstracts) mentioned below (see more details at point #3]), you will see that additional scores like correlation and standard deviation of the error (see point #1]) are as important as the mean error.

4] I think that a very large majority (if not all) radar meteorologists agree with the authors statement "As such, use of traditional Z-S relationships ... cannot be expected to produce accurate results ..." However, a simple Z-S relationship can be used as benchmark. By way of example, in point #4] below, we suggest a possible Z-S to be used to derive as benchmark for the mean and standard deviation of the error as well as for correlation.

5] Please, provide a table with main characteristics of the Ka vertically pointing radar (Antenna type: is it Cassegrain parabolic? 3 or 2 m diameter? Half Power Beam Width?, presence of radome and type, minimum detectable signal in dBm and/or dBZ (including Tx and Rx Losses), Noise Figure, Bandwidth, dynamic range, Tx type: is it a 2 kW traveling wave tube as reported in the radar tutorial? , . . .

[Figure]

6] (see list of suggested references at the end)

Please find below more detailed suggestions/considerations regarding the 6 above-lsited major critical points

1] The mean "error" (actually it is the mean difference between the retrieval and the reference) is presented for 2 events and eventually for a larger pool of 5 events all together (including the first 2 event); in these way counterbalancing error effects are neglected. Please present not only the mean difference, but also the standard deviation of the difference, using hourly (or 3 or 4 hours, I let the authors decide; I see that one problem is also that events are weak . . .) In addition to the standard deviation of the differences, I suggest another score such as the correlation coefficient.

2] As you wrote at page 14, the data from Barrow site is limited. As you wrote, there is a clear need for a larger data set. Cannot you please add some other events with simultaneous radar, snowfall microphysical and snow gage observations?

3] Estimation of solid precipitation at the ground retrieved from remotely sensed measurements is indeed a challenge (see next point #4]), as the authors say in the paper. However, they should emphasize that even in situ point measurements at the ground show sometime huge disagreement depending on wind conditions, absence or single or double fences, . . . A suitable reference for this topic would certainly be the WMO Solid Precipitation Inter-Comparison Experiment (SPICE), which is unfortunately not yet ready (it will soon be) Meanwhile the authors may have a look at the extended abstracts of the WMO TECO and in particular at the extended abstract listed as K3A, O3(1), O3(2), O3(3), O3(4).

http://www.wmo.int/pages/prog/www/IMOP/publications/IOM-125_TECO_2016/TECO-2016_session_3.html

4] The authors emphasize several time the well-known problem (unfortunately, the problem is well known, while the solution is not known!) of non-uniqueness of snowfall

retrievals from remotely sensed observations. [see also point 3]. They could use a single Z-S for the 5 events and show the corresponding scores (e.g. mean and st. dev. of the error, correlation, ...) as benchmark. For instance, they could use the one that is implemented operationally in the Finnish network (Saltikoff et al. 2015) ...

5] Figure 4 and 6: you show a dynamic of 74 dB, from -50 dBZ to +24 dBZ. Is it necessary? Is it reasonable? Is the sensitivity @ 2km range of the Ka Zenith Radar really -50 dBZ?

When you describe Fig. 5, could you please also provide a Table of correlation coefficients among the different curves (if I got it right, 3 values are enough, since LH74 Dendritic and Graupel are perfectly correlated are not they?)

6] SUGGESTED REFERENCES

K3A: Nitu et al. 2016: WMO SPICE: Intercomparison of Instruments and methods for the measurement of Solid Precipitation and Snow on the Ground, Overall results and recommendations, CIMO TECO 2016 extended abstracts, see link above at the end of #3].

O3(1): Roulet et al. 2016: Non-catchment type instruments for snowfall measurement and reporting of precipitation type: General considerations and issues encountered during the WMO CIMO SPICE experiment, and derived recommendations, CIMO TECO 2016 extended abstracts.

O3(2): Lee et al. 2016: Quantitative evaluation of weighing gauges with different wind shields through error modelling, CIMO TECO 2016 extended abstracts.

O3(3): Kochendorfer et al. 2016: Errors, Biases, and Corrections for Weighing Gauge Precipitation Measurements from the WMO Solid Precipitation Intercomparison Experiment, CIMO TECO 2016 extended abstracts.

O3(4): Smith et al. 2016: The WMO SPICE Snow-on-Ground Intercomparison: An Overview of sensor assessment and recommendations on best practices, CIMO TECO

2016 extended abstracts.

Furukawa, K., T. Nio, T. Konishi, R. Oki, T. Masaki, T. Kubota, T. Iguchi, and H. Hanado, 2015: Current status of the dual-frequency precipitation radar on the global precipitation measurement core spacecraft. Proc. SPIE, 9639, 96 390

Iguchi, T., S. Seto, R. Meneghini, N. Yoshida, J. Awaka, M. Le, V. Chandrasekar, and T. Kubota, 2015: GPM/DPR Level-2 Algorithm Theoretical Basis Document. Tech. rep., NASA/JAXA.

Matrasov, S.Y. 2007: Modeling backscatter properties of snowfall at millimiter wavelengths, J. Atmos. Sci, 64, 1727-1736

Saltikoff et al. 2015: Comparison of quantitative snowfall estimates from weather radar, rain gauges and a numerical weather prediction model, BOREAL ENV. RES., 20, 667–678

Speirs et al. 2017: A Comparison between the GPM Dual-Frequency Precipitation Radar and Ground-Based Radar Precipitation Rate Estimates in the SwissAlps and Plateau, DOI: 10.1175/JHM-D-16-0085.1

Minor points

Page 7 Line 24: . . . must then be left large ?? Line 28: . . . rates are based only the lowest few . . .
* * *

---

## Author Comment (AC1) · 27 Apr 2017

April 26, 2017

Author response to peer reviews of 'A variational technique to estimate snowfall rate from coincident radar, snowflake, and fallspeed observations' by S.J. Cooper, N.B. Wood, and T.S. L'Ecuyer

We thank the reviewers for their highly relevant comments and suggestions. We realize now that the original manuscript may have been focused too much on retrieval methodology and first order results. As such, we may have lacked some simple analyses that could have provided more insight into our results. Therefore, we have tried to implement reviewer suggestions, wherever possible, to bridge our retrieval work with some of the more practical issues and techniques concerning the snowfall measurement community. We still feel that the limited number of snow events inhibit robust conclusions of retrieval performance, especially as a function of specific environmental conditions. But we do think these changes better illustrate the potential of our scheme and have improved the paper. Please find below a detailed response to all comments and concerns below. Comments in black are original reviewer comments. **Comments in bold black are our responses to those concerns.** Comments in blue represent the changes in the manuscript in response to specific reviewer concerns.

Changes made in response to major comments include:

1) Inclusion and discussion of 'average differences' for the individual snow events in addition to the original focus on differences in total accumulation over the five events

2) Comparison of hourly NWS snowfall observations with retrieved snowfall rates with correlation coefficients for the April 23 event

3) A quantitative discussion of the impact of observed wind speeds on retrieval results (to the extent possible)

4) Comparison of our snowfall retrieval results with those found from 35 GHz  $Z_e$ -S relationships found in the literature

5) Expanded discussion of Ka-band ARM Zenith Radar and uncertainties associated with snow gauge measurements

**Response to Reviewer 1:**

Major Comment 1) In section 3 (results), you emphasize multiple times the non-uniqueness of snowfall retrievals from radar reflectivities and the issues that it implies: difficulty to assess which combination of retrieval assumptions is truly the best, compensating errors can lead wrong combinations of assumptions to snowfall accumulation in agreement with the reference, etc. The results are presented individually for 2 events and eventually merged into one table showing the total snowfall amount accumulated over five snow events. Given these counterbalancing error effects, I think showing the total snowfall amount error is not very informative and could be misleading. Alternatively, the average error (that you mention p.12, 1.8) and its variability are in my opinion more relevant. I would suggest to add this information in Table 3 and discuss the values obtained in Section 3.4

Response: We agree with the reviewer and have added a discussion of 'average error' for the different retrieval assumption permutations in both the results and conclusions. We also modified Table 4 by adding these average error values. We did not include these values in the original manuscript, in part, because two of the storm events (May 21 and May 26) had snowfall less than ~ 1mm LWE. Small differences between retrieved values and NWS observations in LWE (0.5 mm) would then have a huge impact on average differences and their variability for the five events from a percentage standpoint.

So, instead of reporting the average differences with variability for all five events, we chose to use 'accumulation-weighted average absolute difference'. Essentially, we weighted the 'error' (absolute fractional difference) from each storm proportional to its total snowfall. So, if Storm 1 produced twice as much snowfall as Storm 2, the error of Storm 1 would be weighted twice as much as the error of Storm 2 when calculating our 'average'. The use of 'accumulation-weighted average absolute difference' as a retrieval metric really didn't alter the discussion or conclusions relative to the original manuscript. Of course, for those retrieval cases where results were biased either high or low for all storm cases, this accumulation-weighted average returns the same value as the difference in total accumulation as in the original paper. But for those retrieval scenarios where snowfall was sometimes too high but sometimes too low, this approach gives additional information on retrieval performance.

**Section 3.4 changes**

P13, Line 26: The use of total accumulation over multiple snow events provides a practical metric to evaluate retrieval performance over time. However, it is also important to evaluate the average error for the individual storm events since a retrieval scheme could produce perfect results over multiple storm events while being markedly wrong for each event. Table 4 lists the accumulation-weighted average absolute differences for the individual snow events for different retrieval permutations. We chose this accumulated-weighted approach since two of the storm events had snowfall accumulation less than 1.05 mm liquid equivalent. If we had equally weighted the events, small differences in retrieved snowfall values on the order of 0.5 mm for these events could heavily skew overall average differences.

Table 4 shows accumulation-weighted average absolute differences ranging from 21% to 122% relative to NWS observations for the different retrieval permutations. For our base assumptions (MASC fallspeed, MASC PSD  $\lambda$ , and CloudSat particle model), we found an accumulation-weighted average absolute difference of 36%. This result is similar to the 35% value found with use of the C3VP a priori PSD  $\lambda$ , again demonstrating the limited sensitivity of retrieval results to changes in PSD  $\lambda$  for these events. Two of the smallest average difference values (21% and 27%) were found using hexagonal columns. Again, since columns were very rarely seen during these events, it is likely that such agreement arises from compensating errors from other retrieval assumptions. Similar average difference values (23%, 27%, and 27%) were found for a variety of retrieval combinations based upon a sector plate particle model. Such consistent results identify the sector plate model as a likely habit candidate for a Barrow based snowfall retrieval scheme.

**Section 4 changes:**

P 16, Line 9: This agreement, of course, could result in part from compensating errors in individual storm totals. We also calculated the accumulation-weighted average absolute difference for the individual events and found a value of 36%.

P 16, Line 14: The use of the sector plate model with different fallspeed and PSD assumptions often produced average error values less than 30%, suggesting the sector plate as a candidate for a Barrow-based snowfall scheme. These results are not surprising since both aggregates and sector plates were seen during these snow events.

P 16, Line 21: The accumulation-weighted average absolute difference values for these permutations ranged from 21% to 122%.

P 16, Line 25: Further, this combination had an average difference (28%) for the individual storms that was indeed better than the value found for our base assumptions (36%).

| Table 4: Total retrieved snowfall amounts over five Barrow snow events for designated retrieval assumption combinations.     |
|------------------------------------------------------------------------------------------------------------------------------|
| Nearby NWS observations suggested 16 mm (0.63 inches) of snowfall liquid equivalent. The MASC-derived PSD slope              |
| parameters used for these events are listed in Table 1. The '% Difference' value refers to the difference between the NWS    |
| and retrieved accumulations over the five snow events. The 'Avg % Difference' value refers to the average difference for the |
| individual events.                                                                                                           |

| Particle Model | λ (mm^-1)   | Fallspeed      | Snowfall (mm)           | % Difference | Avg % Difference |  |
|----------------|-------------|----------------|-------------------------|--------------|------------------|--|
| CloudSat       | MASC        | MASC obs       | 13.11                   | -18          | 36               |  |
| CloudSat       | MASC        | Doppler        | 30.35                   | +89          | 89               |  |
| CloudSat       | MASC        | LH74, Aggs Den | 20.60                   | +29          | 38               |  |
| CloudSat       | MASC        | LH74, Graupel  | LH74, Graupel 30.99 +94 |              | 94               |  |
| CloudSat       | MASC        | 1 m/s          | 25.63                   | +60          | 60               |  |
| CloudSat       | Field- C3VP | MASC obs 15.47 |                         | -3           | 35               |  |
| CloudSat       | Field- C3VP | Doppler        | 35.53                   | +122         | 122              |  |
| Sector Plates  | MASC        | MASC obs       | 8.58                    | -46          | 46               |  |
| Sector Plates  | MASC        | Doppler        | 19.74                   | +23          | 23               |  |
| Sector Plates  | MASC        | 1 m/s          | 16.46                   |              | 27               |  |
| Sector Plates  | Field- C3VP | 1 m/s          | 17.20                   | +7           | 27               |  |
| Hex Columns    | MASC        | MASC obs       | 5.79                    | -64          | 64               |  |
| Hex Columns    | MASC        | Doppler        | 13.38                   | -16          | 21               |  |
| Hex Columns    | Field- C3VP | 1 m/s          | 11.94                   | -26          | 28               |  |

Along the same idea, it would be interesting and relevant to analyze the temporal evolution of snowfall accumulation as retrieved using different retrieval assumptions and compare it with the rain gauge measurement, especially for the April 23 event where you use 2 different estimates of the PSD slope parameter for 2 different periods. For instance, a correlation analysis could provide relevant complementary information to evaluate what are the best combinations of retrieval assumptions and to quantify the sensitivity of the retrieval results.

Response: As suggested, we plotted hourly NWS snowfall accumulations against retrieved values for the April 23 snow event for several different retrieval assumption combinations. We introduced a new Figure 7 that visualizes these results. We also found Pearson correlation coefficients for the different curves.

**Section 3.2 changes:**

Page 11. Line 27: In addition to total accumulation, we also considered the temporal distribution of the snowfall during the event. Figure 7 plots NWS hourly snowfall accumulations for the April 23 storm against retrieved values arising from different assumption permutations. Estimates from the CloudSat B8pr-30 particle model (with MASC fallspeed and MASC PSD) agreed reasonably well in trend and magnitude with the NWS hourly observations with a Pearson correlation coefficient of 0.65 and a +7% overestimate of snowfall over the entire event. Retrieval results had a higher correlation with the hourly observations for the first part of the storm (0.92) than the second (0.12) even though differences in subsection snowfall totals were greater for the first section (+11%) than for the second (+1%). However, we do not necessarily expect perfect correlation between the NWS observations and the retrieved values given the differences in location between the snow gauge and KAZR. For example, the NWS snow gauges observed significant snowfall (0.76 mm) over the last two hours of the event even though low KAZR reflectivities suggested perhaps the weakest part of the storm.

The sector plate model (with MASC fallspeed and MASC PSD) produced slightly too little snowfall throughout the course of the event. The hexagonal model (with MASC fallspeed and MASC PSD) produced even less. The three curves for the retrieval scenarios using the MASC fallspeeds in Figure 7 were highly correlated with each other (coefficients of 0.99) given common reflectivity and fallspeed assumptions. These curves had a weaker correlation (coefficients near 0.8) with the Doppler scenario given marked discrepancies in fallspeed as seen in Figure 5.

Figure 7: Hourly NWS snowfall accumulations (mm) plotted against retrieved values for the April 23 event for different retrieval assumption permutations. MASC PSD  $\lambda$  is assumed for all curves.

Major Comment 2) In my opinion, the paragraphs related to fallspeed estimation require some clarifications. How are the fallspeed measured by the MASC inserted into the retrieval scheme? Are you using hourly average as illustrated on Figure 5? How is the fall- speed calculated when multiple particles are present in one MASC image? As you mentioned, Garrett and Yuter 2014 showed that MASC fallspeed measurements are strongly influenced by wind and local turbulence. This will have an impact on the average observed fallspeed but also on its variability. What is the fallspeed variability during the events of interest? You could for instance display this variability on Figure 5 (for MASC and Doppler radar).

Response: In the retrieval scheme, average hourly fallspeeds from the MASC were used to translate retrieved snow water contents (SWC) into snowfall rate for each KAZR vertical profile. So, any SWC within said hour would be multiplied by the same hourly average fallspeed. We added a sentence confirming this in Section 3.1. We agree that better temporal resolution in fallspeed would be ideal but we simply did not have much data for these often weakly precipitating snow events. Many hours of MASC data would contain less than 100 fallspeed observation, so  $\sim 1.5$  minute. Instead of trying to directly correlate observed SWC to nearest fall speed observations for some random particle size, we thought it better to just average over an hour to try to get a representative value. We also agree that looking at variability of fallspeed could also provide some interesting insights into turbulence and sampling issues given a more complete data set.

The MASC images were processed by Dr. Tim Garrett using the same techniques as in Garrett et al., 2012; Garrett and Yuter, 2014; Garrett et al., 2015 (now referenced in Section 2.3). Due to the 'non-unique' nature of the retrieval problem, we did not want to give the appearance of playing with the MASC analysis to get results closer to NWS snow observations. Nor did we want to give the appearance that Dr. Garrett was playing with the data to best match NWS observations given his financial interest in MASC performance. Refining the MASC analysis is certainly a valid research topic, but again the purpose of this paper was more about presenting the methodology of the combined radar, in-situ retrieval with first order results.

**Changes:**

Page 7, Line 13: For the PSD slope parameter a priori guess, the MASC images were processed to quantify maximum particle dimension for each snowflake according to the techniques developed in Garrett et al. (2012), Garrett and Yuter (2014), and Garrett et al. (2015).

Page 9, Line 27: For the MASC fallspeed calculations, retrieved snowfall water contents for a given KAZR reflectivity profile were converted to precipitation rate via their corresponding average hourly fallspeed value.

How was the MASC deployed during the campaign? Was it windshielded? Horizontal wind and turbulence have also a significant impact on the catching ratio of rain gauges. As a rain gauge is used for reference in this study, it would be useful to document the wind conditions during the events and discuss the impact on the results (if available).

Response: The deployment of the MASC at Barrow in Spring 2014 was a test scenario before it was moved to the Oliktok Point site. It was not wind-shielded. We realize this is not ideal for the in-situ measurements. But we believe the data is still well-suited to demonstrate the retrieval technique and provide reasonable first order results and sensitivities. We also agree that windy conditions could cause possible sampling artifacts that could bias both the snow gauge and MASC observations. As such, we have included a discussion of wind conditions (to the extent possible) and their possible influence on our retrieval performance.

Page 6, Line 8: The MASC ideally would be wind-shielded to help reduce sampling errors due to wind and turbulence (Goodison et al., 1998). Given the temporary nature of the deployment at Barrow, however, the MASC was left unshielded during the storm events presented in this work.

Page 8, Line 26: We note that these discrepancies could have been exacerbated, in part, from the unshielded status of the MASC.

**Changes in Section 3.4:**

Page 14, Line 12: In theory, analyses of retrieval performance could be broken down as a function of meteorological conditions. For example, we might expect that the retrievals would match snow gauge observations poorly under windy conditions. Snowfall gauge measurements can suffer significant uncertainties that are a strong function of wind speed (Black, 1956; Larson and Peck, 1974; Goodison et al., 2008; Yang et al., 2005, Wolff et al., 2014; among numerous others). Likewise, sampling artifacts for MASC particle size and fallspeed observations are likely as increased wind speed and turbulence disrupt the settling of snowflakes as they pass through the MASC sampling volume. In practice, we had only a handful of snow events during the Barrow deployment, making a thorough exploration of this topic difficult. For example, retrievals with our base assumptions matched the snow gauge observations better for the slightly less windy April 23 event (2.85 m/s) than the May 15 (4.40 m/s) event. For the April 23 event, however, the hourly snow gauge and retrieval estimates were more highly correlated during the windier first part of the storm (3.96 m/s) than the calmer second part (2.17 m/s). Both events were also complicated by habit uncertainties, making it difficult to identify a clear wind signature without more observations of snow storms.

Perhaps of more interest, the May 21 event had the strongest average winds at 9.41 m/s (21.0 mph), the greatest measured slope parameter, and thus the smallest particles. These results would be consistent with blowing snow composed of shattered ice crystals. If so, we would expect poor retrieval performance as the PSD and habit measurements near the surface would not be representative of the snowfall conditions aloft. The total snowfall accumulation for this high wind event was only 1 mm (LWE), so again it would be unwise to make conclusions on retrieval performance until we get more data.

**Minor comments :**

p2, 119-25: As this paragraph is about spaceborne measurements of snowfall, I would suggest to mention the more recent Global Precipitation Measurement (GPM) mission carrying a dual-frequency precipitation radar. Possible citation: Global Precipitation Measurement Cold Season Precipitation Experiment (GCPEX): For Measurement's Sake, Let It Snow, Skofronick-Jackson, Gail, et al. BAMS 2015. Reference added P3, Line7

p3, 113: I think MASC is the acronym for Multi-Angle Snowflake Camera (not Multiple-Angle Snow Camera). Correction made

p3, 117: You could add a reference for the Precipitation Imaging Package.

**The PIP is a 2nd generation version of the SVI instrument, but there is no specific reference for the PIP.**

Reference for SVI added P3, Line 18

Newman, A. J., Kucera, P. A., and Bliven, L. F.: Presenting the snowflake video imager (SVI), J. Atmos. Ocean. Tech., 26, 167–179, 2009.

p3, 130: we discuss the methodology Correction made

p4, 110: lambda is the PSD slope [...] and D is the particle maximum dimension Correction made

p4, l24: What do you mean by cubic ice dipoles? Ice dipoles arranged on a cubic grid?

Two things are meant here. First, yes, the ice dipoles are arranged on a regular cubic (uniform grid spacing in the x-, y-, and z-dimensions) grid. Second, the mass of each dipole is equal to the mass of a cube of ice whose sides are equal in length to the grid spacing.

p5, l5: I am not sure we can call the SVI a disdrometer, maybe reformulate.

The term 'disdrometer' seems originally to have been used to describe instruments used to measure size distributions of rain drops in particular, but the term is now applied generally to devices measuring the size distributions of hydrometeors, which is what is done by the SVI. Maybe using the term 'video disdrometer' would be a good compromise.

P5, Line 6: DDA scattering properties were used in combination with video disdrometer observations of snow particle size distributions (the Snowflake Video Imager, Newman et al., 2009) to calculate synthetic W-band reflectivities.

P5, 11-14: What assumptions are used on the orientation of the particles in the model?

For the particles from Wood et al. (2015) shown in Figure 1, the particles are assumed to be preferentially oriented with their longest dimension lying nominally in the horizontal plane, with canting angles ranging over +/- 10 degrees, sampled uniformly in the cosine of the angle (this is DDSCAT's standard sampling method for this angle). These orientations were intended to simulated roughly the canting angle distributions found by Matrosov et al. (2005), although their results were for pristine dendritic particles. Further, orientation about the vertical axis were assumed random and sampled uniformly. This is described in the Wood et al. (2015) paper.

p5, 118 (and others): For a publication in an international journal as AMT, I would suggest to convert all measurements in SI units and potentially keep the Imperial units in brackets. Correction made

p5, l28: graupels Correction made

p6, l27: Please clarify why you use this standard deviation value (2 dBZ)

P7, Line 4: We use a  $S_y$  standard deviation value of 2 dBZ for this study for the diagonal matrix elements based upon Hammonds et al. (2014) who quantified the uncertainty in forward modeled radar reflectivity due to assumptions such as snowflake mass-dimensional relationships at Ka-band frequency.

Hammonds, K. D., Mace, G. G., & Matrosov, S. Y. (2014). Characterizing the radar backscatter-cross-section sensitivities of icephase hydrometeor size distributions via a simple scaling of the Clausius-Mossotti factor. *Journal of Applied Meteorology and Climatology*, *53*(12), 2761-2774. DOI: 10.1175/JAMC-D-13-0280.1

p7, 111-18: If I understood correctly, you use MASC-derived PSD slope parameter as a priori guess for the whole profile of KAZR reflectivities. If so, would it make sense to take into account the altitude in the variance of the slope parameter guess (sigma), as the MASC measurement of PSD is less likely to be representative of the real PSD higher in the precipitation column?

Yes, of course, the reviewer is correct. The PSD observed at the surface may not necessarily reflect the PSD aloft that are being observed by the KAZR. However, it should be noted that we were looking at the very bottom of the cloud at ~ 160m above the surface to determine surface snowfall rate. Our assumption would be much worse if we were trying to retrieve vertical profiles of precipitation up through storm top. Along those lines, the authors (UW ones) are currently looking at this exact problem to see if they can relate surface and aloft PSD correlations through coincident surface and in-situ flight data. Although we could have guessed at

an error value for PSD in this work, we chose not to just arbitrarily select a value as it would introduce false information into the retrieval scheme.

p7, l28: are based only on the [...] Correction made

p8, l2: S\_y instead of Sy (subscript) Correction made

p8, 115-16: Please precise the units for V and D. Correction made

p9, 15: Are sector plates and hexagonal columns models also coming from CloudSat DDA simulations? I see a sector plate model on Figure 1 but not a hexagonal column. Please clarify this point.

**Thanks for catching this oversight.**

Changes in Section 3.1

P 9, Line 19: Scattering properties for these pristine habits were derived following the constrained discrete dipole modeling method described by Wood et al. (2015) with particle dimension relationships from Auer & Veal (1970) and particle mass constraints from Mitchell (1996).

**p9, 110: "a priori" is written two times.**

P9, Line 23: For PSD assumptions, we used MASC estimates of slope parameter with uncertainties as listed in Table 1 as introduced through the a priori guess and covariance terms in Eq. 2.

p9, 113: Even though C3VP slope parameter lies within the range of values measured by the MASC, it is almost 4 times larger than the measured value for certain events. I would move this sentence to page 10 after you showed that PSD slop parameter has the least impact on the variability in estimated snowfall accumulation. Suggestion followed

**p9, 129: snow accumulation of Correction made with switch to metric units**

p11, 124-27: More specifically, rimed snowflakes are expected to be more reflective, hence reducing the PSD number density for a fixed slope parameter. It may compensate somewhat the increased snowfall retrieval due to the higher density of individual particles. Is this heavy riming also present in the fallspeeds measured by the MASC and the Doppler radar, compared to non rimed events? If so, I would suggest to mention it in this paragraph. P12, 126: Multi-Angle Snowflake Camera

**Since we have so little data, I would be hesitant to state that we can identify a signature of riming in the few events we have.**

**p14, 15: Remove "vary" correction made**

p14, 111-15: In this regard, you may want to cite the recent work by Praz et al "Solid hydrometeor classification and riming degree estimation from pictures collected with a MASC" (AMTD, 2017) in which the authors developed a method to automatically identify the type of snowflake and the riming degree from MASC images.

We agree that the work described in Praz et al. (2017) would be nearly ideal for our combined radar-MASC retrieval scheme. As mentioned in the manuscript, we are currently working on developing different graupel scattering models with varying degrees of riming. We also are seeing tons and tons of ice at our current deployment in Norway.

**Change is Section 3.3:**

P12, Line 31: Current work by the authors focuses on scattering calculations for rimed particle models assuming different densities. Ideally, these properties could be used with a hydrometeor classification scheme that estimates degree of riming and melt water such as Praz et al. (2017) or similar for better retrieval performance.

Change is Section 4.0:

P17, Line 16: Use of a hydrometeor classification scheme that estimates degree of riming from MASC images such as that developed by Praz et al. (2017) would provide a quantitative manner to link snowflake images with particle scattering properties.

**Response to Reviewer 2:**

1] to complement the Mean Error with (at least one or two) complementary additional score, for instance, the standard deviation of the error and the correlation coefficient among hourly amounts. Indeed, it is interesting and relevant to have an idea of the temporal evolution of the recorded snowfall amounts during a period of time shorter than the whole event.

1 cont.] The mean "error" (actually it is the mean difference between the retrieval and the reference) is presented for 2 events and eventually for a larger pool of 5 events all together (including the first 2 event); in these way counterbalancing error effects are neglected. Please present not only the mean difference, but also the standard deviation of the difference, using hourly (or 3 or 4 hours, I let the authors decide; I see that one problem is also that events are weak . . .) In addition to the standard deviation of the differences, I suggest another score such as the correlation coefficient.

Response: We agree with the reviewer and have added a discussion of 'average error' for the different retrieval assumption permutations in both the results and conclusions. We also modified Table 4 by adding these average error values. We did not include these values in the original manuscript, in part, because two of the storm events (May 21 and May 26) had snowfall less than ~ 1mm LWE. Small differences between retrieved values and NWS observations in LWE (0.5 mm) would then have a huge impact on average differences and their variability for the five events from a percentage standpoint.

So, instead of reporting the average differences with variability for all five events, we chose to use 'accumulation-weighted average absolute difference'. Essentially, we weighted the 'error' (absolute fractional difference) from each storm proportional to its total snowfall. So, if Storm 1 produced twice as much snowfall as Storm 2, the error of Storm 1 would be weighted twice as much as the error of Storm 2 when calculating our 'average'. The use of 'accumulation-weighted average absolute difference' as a retrieval metric really didn't alter the discussion or conclusions relative to the original manuscript. Of course, for those retrieval cases where results were biased either high or low for all storm cases, this accumulation-weighted average returns the same value as the difference in total accumulation as in the original paper. But for those retrieval scenarios where snowfall was sometimes too high but sometimes too low, this approach gives additional information on retrieval performance.

**Section 3.4 changes**

P13, Line 26: The use of total accumulation over multiple snow events provides a practical metric to evaluate retrieval performance over time. However, it is also important to evaluate the average error for the individual storm events since a retrieval scheme could produce perfect results over multiple storm events while being markedly wrong for each event. Table 4 lists the accumulation-weighted average absolute differences for the individual snow events for different retrieval permutations. We chose this accumulated-weighted approach since two of the storm events had snowfall accumulation less than 1.05 mm liquid equivalent. If we had equally weighted the events, small differences in retrieved snowfall values on the order of 0.5 mm for these events could heavily skew overall average differences.

Table 4 shows accumulation-weighted average absolute differences ranging from 21% to 122% relative to NWS observations for the different retrieval permutations. For our base assumptions (MASC fallspeed, MASC PSD  $\lambda$ , and CloudSat particle model), we found an accumulation-weighted average absolute difference of 36%. This result is similar to the 35% value found with use of the C3VP a priori PSD  $\lambda$ , again demonstrating the limited sensitivity of

retrieval results to changes in PSD  $\lambda$  for these events. Two of the smallest average difference values (21% and 27%) were found using hexagonal columns. Again, since columns were very rarely seen during these events, it is likely that such agreement arises from compensating errors from other retrieval assumptions. Similar average difference values (23%, 27%, and 27%) were found for a variety of retrieval combinations based upon a sector plate particle model. Such consistent results identify the sector plate model as a likely habit candidate for a Barrow based snowfall retrieval scheme.

**Section 4 changes:**

P 16, Line 9: This agreement, of course, could result in part from compensating errors in individual storm totals. We also calculated the accumulation-weighted average absolute difference for the individual events and found a value of 36%.

P 16, Line 14: The use of the sector plate model with different fallspeed and PSD assumptions often produced average error values less than 30%, suggesting the sector plate as a candidate for a Barrow-based snowfall scheme. These results are not surprising since both aggregates and sector plates were seen during these snow events.

P 16, Line 21: The accumulation-weighted average absolute difference values for these permutations ranged from 21% to 122%.

P 16, Line 25: Further, this combination had an average difference (28%) for the individual storms that was indeed better than the value found for our base assumptions (36%).

**Modified Table 4:**

Table 4: Total retrieved snowfall amounts over five Barrow snow events for designated retrieval assumption combinations. Nearby NWS observations suggested 16 mm (0.63 inches) of snowfall liquid equivalent. The MASC-derived PSD slope parameters used for these events are listed in Table 1. The '% Difference' value refers to the difference between the NWS and retrieved accumulations over the five snow events. The 'Avg % Difference' value refers to the average difference for the individual events.

| Particle Model | λ (mm^-1)   | Fallspeed      | Snowfall (mm) | % Difference | Avg % Difference |
|----------------|-------------|----------------|---------------|--------------|------------------|
| CloudSat       | MASC        | MASC obs       | 13.11         | -18          | 36               |
| CloudSat       | MASC        | Doppler        | 30.35         | +89          | 89               |
| CloudSat       | MASC        | LH74, Aggs Den | 20.60         | +29          | 38               |
| CloudSat       | MASC        | LH74, Graupel  | 30.99         | +94          | 94               |
| CloudSat       | MASC        | 1 m/s          | 25.63         | +60          | 60               |
| CloudSat       | Field- C3VP | MASC obs       | 15.47         | -3           | 35               |
| CloudSat       | Field- C3VP | Doppler        | 35.53         | +122         | 122              |
| Sector Plates  | MASC        | MASC obs       | 8.58          | -46          | 46               |
| Sector Plates  | MASC        | Doppler        | 19.74         | +23          | 23               |
| Sector Plates  | MASC        | 1 m/s          | 16.46         | +3           | 27               |
| Sector Plates  | Field- C3VP | 1 m/s          | 17.20         | +7           | 27               |
| Hex Columns    | MASC        | MASC obs       | 5.79          | -64          | 64               |
| Hex Columns    | MASC        | Doppler        | 13.38         | -16          | 21               |
| Hex Columns    | Field- C3VP | 1 m/s          | 11.94         | -26          | 28               |

Response: We also looked at the temporal evolution of the storm, i.e. compared hourly NWS snowfall accumulations against retrieved values for the April 23 snow event for several different retrieval assumption combinations. We introduced a new Figure 7 that visualizes these results. We also found Pearson correlation coefficients for the different curves.

**Section 3.2 changes:**

Page 11. Line 27: In addition to total accumulation, we also considered the temporal distribution of the snowfall during the event. Figure 7 plots NWS hourly snowfall accumulations for the April 23 storm against retrieved values arising from different assumption permutations. Estimates from the CloudSat B8pr-30 particle model (with MASC fallspeed and MASC PSD) agreed reasonably well in trend and magnitude with the NWS hourly observations with a Pearson correlation coefficient of 0.65 and a +7% overestimate of snowfall over the entire event. Retrieval results had a higher correlation with the hourly observations for the first part of the storm (0.92) than the second (0.12) even though differences in subsection snowfall totals were greater for the first section (+11%) than for the second (+11%). However, we do not necessarily expect perfect correlation between the NWS observations and the retrieved values given the differences in location between the snow gauge and KAZR. For example, the NWS snow gauges observed significant snowfall (0.76 mm) over the last two hours of the event even though low KAZR reflectivities suggested perhaps the weakest part of the storm.

The sector plate model (with MASC fallspeed and MASC PSD) produced slightly too little snowfall throughout the course of the event. The hexagonal model (with MASC fallspeed and MASC PSD) produced even less. The three curves for the retrieval scenarios using the MASC fallspeeds in Figure 7 were highly correlated with each other (coefficients of 0.99) given common reflectivity and fallspeed assumptions. These curves had a weaker correlation (coefficients near 0.8) with the Doppler scenario given marked discrepancies in fallspeed as seen in Figure 5.

Figure 7: Hourly NWS snowfall accumulations (mm) plotted against retrieved values for the April 23 event for different retrieval assumption permutations. MASC PSD  $\lambda$  is assumed for all curves.

2] Three events out of five are indeed weak. Would it be possible to please add more events?

2 cont..] As you wrote at page 14, the data from Barrow site is limited. As you wrote, there is a clear need for a larger data set. Cannot you please add some other events with simultaneous radar, snowfall microphysical and snow gage observations?

Response: Unfortunately, the required combination of MASC, KAZR, and snow gauge observations was only available for a limited time at Barrow. The MASC was deployed in Spring 2014 when we had the events described in the paper. Our first good event was April 23 and precipitation events soon after often contained rain or mixed conditions. From a gauge standpoint, there is no way to differentiate between rain and snow. The MASC was then moved to the Oliktok Point long term deployment where it was not made operational until Fall 2015. The authors are unaware of any available snowfall gauge measurements at this new site. However, we still feel we had enough data from Barrow to illustrate the optimal-estimation technique and present preliminary results and sensitivities. As mentioned in the paper, we do have such a deployment in Norway this winter and in Sweden next winter. We expect to apply the retrieval method presented here to the more comprehensive data sets from those sites to more fully examine the scientific concerns brought up by both

**reviewers. For example, this should provide us the opportunity to apply the statistical analyses as a function of meteorological conditions that both reviewers suggested.**

3] Would you please supply more information regarding the reference in situ observations? Furthermore, please write a few sentences characterizing the mountainous site at hand. In addition to snowfall amounts and water equivalent, do you have information regarding the wind intensity (and direction)? All considered, after having read the interesting the cited paper by Wolff et al. (2014) and other "grey" literature (ex- tended abstracts) mentioned below (see more details at point #3]), you will see that additional scores like correlation and standard deviation of the error (see point #1]) are as important as the mean error.

3 cont..] Estimation of solid precipitation at the ground retrieved from remotely sensed mea- surements is indeed a challenge (see next point #4]), as the authors say in the paper. However, they should emphasize that even in situ point measurements at the ground show sometime huge disagreement depending on wind conditions, absence or single or double fences, ... A suitable reference for this topic would certainly be the WMO Solid Precipitation Inter-Comparison Experiment (SPICE), which is unfortunately not yet ready (it will soon be) Meanwhile the authors may have a look at the extended abstracts of the WMO TECO and in particular at the extended abstract listed as K3A, O3(1), O3(2), O3(3), O3(4).

Response: The deployment of the MASC at Barrow in Spring 2014 was a test scenario before it was moved to the Oliktok Point site. It was not wind-shielded. We realize this is not ideal for the in-situ measurements. But we believe the data is still well-suited to demonstrate the retrieval technique and provide reasonable first order results and sensitivities. We also agree that windy conditions could cause possible sampling artifacts that could bias both the snow gauge and MASC observations. As such, we have included a discussion of wind conditions (to the extent possible) and their possible influence on our retrieval performance.

Page 6, Line 8: The MASC ideally would be wind-shielded to help reduce sampling errors due to wind and turbulence (Goodison et al., 1998). Given the temporary nature of the deployment at Barrow, however, the MASC was left unshielded during the storm events presented in this work.

Page 8, Line 26: We note that these discrepancies could have been exacerbated, in part, from the unshielded status of the MASC.

Changes in Section 3.4:

Page 14, Line 12: In theory, analyses of retrieval performance could be broken down as a function of meteorological conditions. For example, we might expect that the retrievals would match snow gauge observations poorly under windy conditions. Snowfall gauge measurements can suffer significant uncertainties that are a strong function of wind speed (Black, 1956; Larson and Peck, 1974; Goodison et al., 2008; Yang et al., 2005, Wolff et al., 2014; among numerous others). Likewise, sampling artifacts for MASC particle size and fallspeed observations are likely as increased wind speed and turbulence disrupt the settling of snowflakes as they pass through the MASC sampling volume. In practice, we had only a handful of snow events during the Barrow deployment, making a thorough exploration of this topic difficult. For example, retrievals with our base assumptions matched the snow gauge observations better for the slightly less windy April 23 event (2.85 m/s) than the May 15 (4.40 m/s) event. For the April 23 event, however, the hourly snow gauge and retrieval estimates were more highly correlated during the windier first part of the storm (3.96 m/s) than the calmer second part (2.17 m/s). Both events were also complicated by habit uncertainties, making it difficult to identify a clear wind signature without more observations of snow storms.

Perhaps of more interest, the May 21 event had the strongest average winds at 9.41 m/s (21.0 mph), the greatest measured slope parameter, and thus the smallest particles. These results would be consistent with blowing snow composed of shattered ice crystals. If so, we would expect poor retrieval performance as the PSD and habit measurements near the surface would not be representative of the snowfall conditions aloft. The total snowfall accumulation for this high wind event was only 1 mm (LWE), so again it would be unwise to make conclusions on retrieval performance until we get more data.

Response: We also added numerous references and a quick discussion of snow gauge uncertainties in Section 31. and reiterated its importance in the conclusions.

**Changes in Section 3.1**

P10, Line 3: Furthermore, the accurate measurement of snowfall amount from precipitation gauges is a challenging research topic in its own right (Goodison et al., 1998; Rasmussen et al., 2012). Numerous studies over the past century have sought to quantify uncertainties in these measurements and then provide a correction factor for environmental conditions (e.g. Black, 1956; Larson and Peck, 1974; Yang et al., 1995; Yang et al., 2005). During windy and turbulent conditions, for example, the gauges likely will underestimate snowfall amount as the snowflakes cannot settle into the gauge opening. Overall, collection efficiency is a complex function of instrument design, shielding, wind, turbulence, temperature, and topography among other factors (Folland, 1988; Goodison et al., 1998; Rasmussen et al., 2012, Theriault et al., 2015) and can result in snowfall uncertainties on the order of 25-50% over the course of a winter season (Wolff et al., 2014). As such, we understand that the reported NWS snowfall values cannot be taken at absolute face value.

4] I think that a very large majority (if not all) radar meteorologists agree with the au- thors statement "As such, use of traditional Z-S relationships . . . cannot be expected to produce accurate results . . ." However, a simple Z-S relationship can be used as benchmark. By way of example, in point #4] below, we suggest a possible Z-S to be used to derive as benchmark for the mean and standard deviation of the error as well as for correlation.

4 cont...] The authors emphasize several time the well-known problem (unfortunately, the problem is well known, while the solution is not known!) of non-uniqueness of snowfall retrievals from remotely sensed observations. [see also point 3]. They could use a single Z-S for the 5 events and show the corresponding scores (e.g. mean and st. dev. of the error, correlation, . . .) as benchmark. For instance, they could use the one that is implemented operationally in the Finnish network (Saltikoff et al. 2015) . . .

Response: We agree with the reviewer comments. We implemented two different  $Z_e$ -S relationships for 35 GHz, one from Kulie and Bennartz (2009) and one from Matrosov (2007). We did not implement the Saltikoff et al. (2015) as it seemed to be based upon operational weather radar with a different frequency than the KAZR observations. (Although, we certainly agree in spirit with the Saltikoff et al. assessment that the 'variability of the microphysical properties of snow is so large that a perfect operational solution for all situations may be impossible to reach.') We picked these two different schemes because they were based upon different scattering solution techniques. And we did not endlessly search for schemes that would provide us the answer we were looking for. We picked those two schemes first and used whatever came out of them.

We included a discussion of both the differences in total accumulation for the five events and the accumulated weighted average absolute difference for the events for these  $Z_e$ -S schemes (this information is in a new Table 5). KB09 and M07 schemes overestimated and underestimated snowfall, respectively, relative to NWS observations but still fell within the range of values found with our OE schemes. Both  $Z_e$ -S relationships had 'average errors' near 50% for the events. We also calculated the correlation coefficient between hourly Z-S estimates and NWS obs for the April 23 event.

**Changes in Section 3.4:**

P14, Line 31: Finally, we contrast snowfall estimates found with our variational approach with those found from two different 35 GHz reflectivity- snowfall relationships. Our use of in-situ observations from the Barrow event, in theory, should produce more accurate results than those derived from these parameterizations. Kulie and Bennartz (2009, hereafter KB09) estimated snowfall as in Eq. 7,

**$Z_e = 24.04S^{1.51}$ (7)**

where they assumed a three-bullet rosette particle model (Liu 2008b) with scattering particles derived from DDA. S is snowfall in mm hr-1. Matrosov (2007a, hereafter M07) assumed aggregate snowflakes with scattering properties modeled from spheres and T-matrix theory to parameterize snowfall through Eq. 8.

**$Z_e = 56.00S^{1.2}$ (8)**

These equations were derived for vertically pointing radar and dry snowfall conditions that lack significant attenuation at 35 GHz.

Table 5 lists total estimated snowfall and accumulation-weighted average absolute differences for the KB09 and M07 schemes for the five Barrow storm events. KB09 found a total accumulation with +37% difference relative to NWS observations with an average error of 50% for the individual events. M07 found a total accumulation with -48% difference relative to NWS observations with an average error of 49% for the individual events. These values fall well within the range of results presented in Table 4, although they are slightly larger than those found from the better case optimal-estimation retrieval scenarios. Likewise, the correlation coefficient between hourly  $Z_e$ -S estimates and NWS observations (0.41) was slightly worse than that found between the base OE scheme and NWS observations (0.65). Again, it is hoped the continued refinement of our combined radar, in-situ approach will produce results consistently more accurate than such  $Z_e$ -S relationships.

**Changes in Section 4:**

P16, Line17: These better case retrieval scenarios matched NWS observations more closely than two 35 GHz  $Z_e$ -S relationships that had average errors near 50% for the Barrow events.

| Table 5: Retrieved snowfall amounts as derived from two different Ze-S relationships for April 23, May 15, and all sno | w  |
|------------------------------------------------------------------------------------------------------------------------|----|
| events seen at Barrow. These results are contrasted with those found using our base optimal-estimation scheme and NW   | 'S |
| observations. Difference values are defined as in Table 4.                                                             |    |

|                    | April 23 |        | May 15   |        | All Events |        |            |
|--------------------|----------|--------|----------|--------|------------|--------|------------|
|                    | LWE (mm) | % Diff | LWE (mm) | % Diff | LWE (mm)   | % Diff | Avg % Diff |
| NWS                | 4.57     |        | 7.62     | -      | 16.0       | -      | -          |
| Base Scheme | 4.88     | +7     | 4.34     | -43    | 13.11      | -18    | 36         |
| Kulie 09           | 8.71     | +90    | 6.63     | -13    | 21.97      | +37    | 50         |
| Matrosov 07        | 3.53     | -23    | 2.68     | -65    | 8.37       | -48    | 49         |

5] Please, provide a table with main characteristics of the Ka vertically pointing radar (Antenna type: is it Cassegrain parabolic? 3 or 2 m diameter? Half Power Beam Width?, presence of radome and type, minimum detectable signal in dBm and/or dBZ (including Tx and Rx Losses), Noise Figure, Bandwidth, dynamic range, Tx type: is it a 2 kW traveling wave tube as reported in the radar tutorial?, ...

Response: We certainly agree that more information on the KAZR should have been included. We have included a few sentences discussing KAZR attributes as well as a reference for the KAZR written that other peer-reviewed studies have referenced. All relevant radar technical specifications are in that article. We also included a quick calculation documenting that the KAZR minimum detectable signal is sufficient to examine the snowfall studies observed during these events.

**Changes in Section 2.3:**

P6, Line13: Given lack of W-band radar measurements during this time period, we use Ka-band ARM Zenith radar (KAZR) general mode observations for this study. The KAZR is vertically-pointing radar that measures reflectivity, vertical velocity, and spectral width at 34.8 GHz with a resolution of 30 m from near surface up to about 20 km (Bharadway et al., 2013). The radar had a 2m antennae with 0.31 degree beam width. KAZR systems have a minimum detectable signal in general mode near -20 dBZ to -25 dBZ dependent upon target range (Feng et al., 2014; Chandra et al., 2015). Such performance is sufficient for these studies given we need reflectivities near -10 dBZ during the entire day to generate measureable snowfall (0.01 inches) as reported by the National Weather Service.

Bharadwaj, N., A. Lindenmaier, K. B. Widener, K. L. Johnson, and V. Venkatesh, 2013: Ka-band ARM zenith profiling radar (KAZR) network for climate study. 36th Conf. on Radar Meteorology, Breckenridge, CO, Amer. Meteor.Soc.,14A.8.[https://ams.confex.com/ams/36Radar/webprogram/Manuscript/Paper228620/14A8\_ams\_radconf\_kazr.pdf.]

Chandra, A., Zhang, C., Kollias, P., Matrosov, S., and Szyrmer, W.: Automated rain rate estimates using the Ka-band ARM zenith radar (KAZR), Atmos. Meas. Tech., 8, 3685-3699, doi:10.5194/amt-8-3685-2015, 2015.

Feng, Z., S. A. McFarlane, C. Schumacher, S. Ellis, J. Comstock, and N. Bharadwaj, 2014: Constructing a merged cloud– precipitation radar dataset for tropical convective clouds during the DYNAMO/AMIE experiment at Addu Atoll. *J. Atmos. Oceanic Technol.*, **31**, 1021–1042, doi:10.1175/JTECH-D-13-00132.1.

5 cont...] Figure 4 and 6: you show a dynamic of 74 dB, from -50 dBZ to +24 dBZ. Is it necessary? Is it reasonable? Is the sensitivity @ 2km range of the Ka Zenith Radar really -50 dBZ?

Response: The dBZ range of this figure was selected entirely for presentation purposes given our color scheme. The orange and red parts seem to coincide reasonably with moderate snowfall rates. The green and blue show clouds which allows a viewer to understand the structure of storm. We do not mean to imply a KAZR sensitivity down to the -50 dBZ.

Change in Figure 6 caption: Areas of orange and red indicate likely periods with snowfall.

When you describe Fig. 5, could you please also provide a Table of correlation coefficients among the different curves (if I got it right, 3 values are enough, since LH74 Dendritic and Graupel are perfectly correlated are not they?)

Response: We added the correlation coefficient information in the text Section 2.3 (we already have numerous tables). The MASC fallspeeds were most closely correlated with the LH74 parameterizations with a value of 0.66. The MASC and KAZR Doppler observations were actually negatively correlated with a coefficient of - 0.34. The LH parameterizations indeed were perfectly correlated with each other and had a -0.23 correlation coefficient with the KAZR.

Changes in Section 2.3

P8, Line 25: Indeed, the hourly MASC and KAZR Doppler fallspeeds had a Pearson correlation coefficient of - 0.34.

P9, Line 5: However, the hourly MASC observations still had a higher correlation coefficient with the LH74 schemes (0.66) than with the Doppler fallspeeds (-0.34).

**6] SUGGESTED REFERENCES**

**Response: We added 10 new references related to (6) relating to the study of snow gauge measurement uncertainties.**

References added:

Black, R.F., Precipitation at Barrow, Alaska, greater than recorded, EOS Trans. AGU, 35(2), 203-207, 1954.

Folland, C., 1988. Numerical models of the raingauge exposure problem, field experiments and an improved collector design. Q. J. Roy. Meteor. Soc. 114 (484), 1485-1516.

Goodison, B. E., P. Y. T. Louie, and D. Yang, 1998: WMO solid precipitation measurement intercomparison. WMO Instruments and Observing Methods Rep. 67, WMO/ TD-872, 212 pp.

Larson, L.W., and E.L. Peck (1974), Accuracy of Precipitation Measurements for Hydrologic Modeling, *Water Resources Research*, 10(4), 857-863.

Rasmussen R, Baker B, Kochendorfer J, Meyers T, Landolt S, Fischer AP, Black J, Thériault JM, Kucera P, Gochis D, Smith C, Nitu R, Hall M, Ikeda K, Gutmann E (2012) How well are we measuring snow: the NOAA/FAA/NCAR winter precipitation test bed. B Am Meteorol Soc 93:811–829. doi:10.1175/ BAMS-D-11-00052.1

Saltikoff E., Lopez P., Taskinen A. & Pulkkinen S. 2015: Comparison of quantitative snowfall estimates from weather radar, rain gauges and a numerical weather prediction model. *Boreal Env. Res.* 20: 667–678.

Speirs, P., M. Gabella, and A. Berne (2017), A Comparison between the GPM Dual-Frequency Precipitation Radar and Ground-Based Radar Precipitation Rate Estimates in the Swiss Alps and Plateau, *J. Hydrometeor*, **18**, 1247-1269.

Thériault, J.M., R. Rasmussen, E. Petro, J.Y. Trepanier, M. Colli, and L.D. Lanza, Impact of Wind Direction, Wind Speed, and Particle Characteristics on the Collection Efficiency of the Double Fence Intercomparison Reference, *J. Applied Meteor. and Climate*, (54), 1918-1930, DOI: http://dx.doi.org/10.1175/JAMC-D-15-0034.1

Yang, D, B.E. Goodison, J.R. Metcalfe, V. S. Golubev, R. Bates, T. Pangburn, and C.L. Hanson (1995) Accuracy of NWS 8 Standard Nonrecording Precipitation Gauge: Results and Application of WMO Intercomparison, *J. Atmos. Ocean. Tech.*, (15), 54-69.

Yang, D. D.L. Kane, Z. Zhang, B. Goodison (2005), Bias corrections of long-term (1973-2004) daily precipitation data over the northern regions, *Geophys Res. Lett.*, 32(19), 1-5, DOI: 10.1029/2005GL024057.

**Minor comments:**

Page 7 Line 24: ... must then be left large ?? 'allowed to vary over several orders of magnitude'

Line 28: ... rates are based only the lowest few . . Eliminated 'few'

---

## Author Response (AR2)

June 11, 2017

Author response to peer reviews of 'A variational technique to estimate snowfall rate from coincident radar, snowflake, and fallspeed observations' by S.J. Cooper, N.B. Wood, and T.S. L'Ecuyer

We again thank the reviewers for their comments. We made the minor revisions as suggested in the Referee Report from Reviewer 1. We also added a sentence in the acknowledgments thanking the reviewers for their comments throughout the review process. These suggestions clearly improved our manuscript and will prove valuable in evaluating retrieval performance as we move forward with this line of research. In terms of the larger issues (e.g. 'variability in variability' and additional figures) discussed in the most recent reviews, we stand by our previous comments in our first author response.

Minor corrections:

- p6, l8: "The MASC ideally should be wind-shielded [...]". As far as I know, there is no study investigating MASC (or similar instrument) performance with or without a windshield so I would temper this statement.

Response:  We removed this sentence as the reviewer is correct.  There have been no quantitative studies of the impact of wind-shielding on MASC results.  We do briefly discuss the impact of wind speed, turbulence, and shielding on MASC results in a qualitative manner in Section 2.3 and will leave it at that.

- p12, l9: I would suggest to write "the hexagonal column model" to match the legend of Figure 7.

Response: correction made